# Density-Guided Robust Counterfactual Explanations on Tabular Data under Model Multiplicity

**Jun Tan** [* 1]  **Qing Guo** [* 1]  **Zicheng Xu** [1]  **Jinglin Li** [1]  **QI Fang** [1]  **Ning Gui** [1]

## Abstract

Counterfactual explanations (CEs) are essential for actionable recourse, yet their reliability is often compromised in low-density regions, where classifiers exhibit high variance. Unlike existing methods that rely on expensive ensemble intersections to define stability, we propose *DensityFlow*, a generative framework that constructs robust CEs by adhering to the high-confidence data manifold. Specifically, we model the counterfactual generation as continuous-time dynamics parameterized by Neural ODE, guided by a differentiable density score to actively avoid uncertain, low-density areas. This density score is learned via Noise Contrastive Estimation, effectively leveraging a $(K+1)$-way discriminator to estimate density ratios. For black-box settings, we introduce a local proxy distillation mechanism that aligns a lightweight surrogate with the target model strictly within the trajectory of CE generation, enabling efficient gradient-based optimization with minimal queries. Experiments demonstrate that *DensityFlow* achieves superior validity under model multiplicity while significantly reducing query costs compared to ensemble-based baselines. Our implementation is available at https://github.com/G-AILab/DensityFlow.

## 1. Introduction

Counterfactual explanations (CEs) identify the minimal or most plausible perturbations required to alter a model's prediction for a specific instance to a desired outcome. Serving as a crucial mechanism for algorithmic recourse and transparency, CEs are indispensable in high-stakes decision-

---
[*]Equal contribution  [1]School of Computer Science and Engineering, Central South University, Changsha, China. Correspondence to: QI Fang <csqifang@csu.edu.cn>, Ning Gui <ning-gui@gmail.com>.

*Proceedings of the 43rd International Conference on Machine Learning*, Seoul, South Korea. PMLR 306, 2026. Copyright 2026 by the author(s).

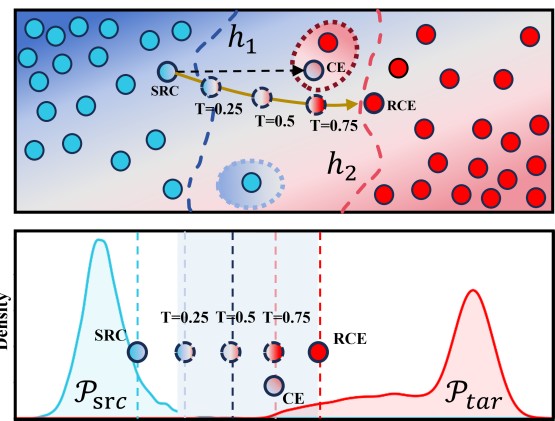

*Figure 1.* Model disagreement in low-density regions along counterfactual trajectories. **Upper**: Plausible classifiers ($h_1$, $h_2$) may induce different decision boundaries in regions with limited data support, causing CEs to pass through areas where validity does not generalize across classifiers. **Lower**: The standard CE tends to fall into the tail of the $\mathcal{P}_{\text{tar}}$. RCE transforms to the high-density region of $\mathcal{P}_{\text{tar}}$, avoiding the uncertainty of the distribution tail.

making scenarios, e.g., risk assessment (Karimi et al., 2022). Formally, given a classifier $h : \mathcal{X} \to \mathcal{Y}$, a target class $y^*$, and a query instance $\boldsymbol{x} \in \mathcal{X}$ where $h(\boldsymbol{x}) \neq y^*$, a CE is defined as a perturbed instance $\boldsymbol{x}' \in \mathcal{X}$ such that $h(\boldsymbol{x}') = y^*$, minimizing a specific cost function $c(\boldsymbol{x}, \boldsymbol{x}')$.

Recently, generative models have emerged as a powerful paradigm in this domain, leveraging the capacity to model complex distributions to enable the rapid conditional generation of counterfactuals. From a theoretical perspective, this process can be framed as a constrained optimal transport problem, aiming to transport a query instance $\boldsymbol{x}$ from the source distribution $\mathcal{P}_{\text{src}}$ to the target distribution $\mathcal{P}_{\text{tar}}$ with minimal cost. Various architectures have been proposed to instantiate this framework, including VAE-based (Pawelczyk et al., 2020b), diffusion-based (Schrodi et al., 2024), and flow-based (Duong et al., 2023). However, by indiscriminately modeling the entire distribution, these methods remain vulnerable to low-confidence samples, which distort the learned manifold and degrade robustness.

Fig. 1 illustrates the inherent risks of counterfactual generation in low-density regions. Since the data distribution is sparse between classes (Upper), the decision boundaries of

plausible classifiers $(h_1, h_2)$ often diverge significantly in these areas (the Rashomon effect(Breiman, 2001b)). Standard CEs, driven by distance minimization, tend to settle in the long-tail regions of the target distribution $\mathcal{P}_{\text{tar}}$ (Lower), where model consensus is weak. Thus, achieving robust counterfactuals (RCE) requires guiding the generation toward high-density regions. Although some non-generative methods address this by enforcing consensus across model ensembles (Jiang et al., 2024a; Stępka et al., 2025), they are bottlenecked by high computational costs and poor scalability, often relying on heuristic constraints that do not fundamentally model the data distribution.

For generative RCEs, capturing the underlying data manifold is essential to ensure transport paths remain within reliable, high-density regions. Motivated by this, we introduce **DensityFlow**, a continuous-flow generator learned under class-wise density guidance. We quantify density by reformulating the $K$-class problem as a $(K+1)$-way discrimination task against uniform noise. Rather than learning density in isolation, we learn the density signal jointly with the predictive validity. This coupling prevents the model from overfitting to sparse outliers, yielding a smoother density landscape. Finally, this robust score guides the flow dynamics, steering trajectories through high-confidence regions while minimizing transport costs. Our contributions are as follows:

**Density-guided Neural ODE Framework.** We propose DensityFlow, a generative framework that ensures robust counterfactuals by constraining the generation process to the high-density data manifold. We implement this via a Density-guided Neural ODE, where the continuous flow dynamics are explicitly steered by a learned density potential to avoid uncertain regions.

**Robust Density Estimation via NCE.** We introduce a $(K+1)$-way formulation that leverages Noise Contrastive Estimation to learn a differentiable density score alongside the classification task. We theoretically prove that this score serves as an effective proxy for sample density, and empirically show that joint optimization with validity prevents overfitting to outliers.

**Trajectory-Aware Local Distillation.** For black-box settings, we design a query-efficient distillation strategy that aligns a lightweight surrogate with the target model only within the regions visited during generation. This approach minimizes redundant queries while maintaining high gradient fidelity in the trusted neighborhood.

Extensive experiments on synthetic and real-world datasets confirm that DensityFlow achieves state-of-the-art robustness and validity under model multiplicity, while requiring significantly fewer queries than existing methods.

**Conflict of interest disclosure:** No conflict of interest.

## 2. Related Work

**Counterfactual Explanations**, often framed as Algorithmic Recourse (Ustun et al., 2019), have become a cornerstone of explainable AI (Karimi et al., 2022). Early works predominantly adopted a per-sample optimization paradigm, iteratively perturbing instances in the input space via gradient descent (Wachter et al., 2017; Mothilal et al., 2020), constrained optimization (Kanamori et al., 2020), or Mixed-Integer Programming (Parmentier & Vidal, 2021).

However, these methods often neglect the underlying data manifold, leading to unrealistic or out-of-distribution explanations. Consequently, recent research has pivoted towards generative models—including VAEs (Pawelczyk et al., 2020b; Panagiotou et al., 2024), diffusion models (Jeanneret et al., 2022; Madaan & Bedathur, 2023; Schrodi et al., 2024), and normalizing flows (Dombrowski et al., 2021; Duong et al., 2023; Wielopolski et al., 2024)—to leverage learned data priors. By performing search or generation within a latent space, these approaches better ensure that the resulting CEs satisfy both feasibility and plausibility.

**Robust Counterfactual Explanations** are typically categorized into four perspectives: model changes, model multiplicity (MM), noisy execution, and input changes (Jiang et al., 2024b). Given our focus, we restrict our discussion to MM-based robustness.

Early research often formulated MM-robustness as a constrained optimization problem, employing Mixed-Integer Linear Programming (MILP) to enforce ensemble consensus (Leofante et al., 2023). To handle heterogeneous architectures, subsequent works shifted toward probabilistic frameworks, using stability metrics (Hamman et al., 2024) or entropic risk (Noorani et al., 2025). In practical black-box settings where model internals are inaccessible (Guidotti, 2024), existing methods rely on rule-based consensus (Jiang et al., 2024a) or stochastic retraining (Stępka et al., 2025). However, these iterative methods suffer from prohibitive query costs. While some methods implicitly consider data support via prototype retrieval (e.g., FACE (Poyiadzi et al., 2020)) or VAE latents (Pawelczyk et al., 2020a), they fail to explicitly utilize differentiable density scores to steer generation, leaving CEs vulnerable to low-density outliers—a gap that our work addresses.

## 3. Problem Statement

### 3.1. Preliminaries and Density Definition

**Notations.** Consider a supervised classification task on an input space $\mathcal{X} \subseteq \mathbb{R}^d$ and a label space $\mathcal{Y} = \{1, \ldots, K\}$. Let $h : \mathcal{X} \to \mathcal{Y}$ denote a trained classifier which we aim to explain. Given a query instance $\boldsymbol{x} \in \mathcal{X}$ with the original predicted label $y = h(\boldsymbol{x})$, the goal of Counterfactual Expla-

nations (CEs) is to find a perturbed instance $\boldsymbol{x}'$ such that the prediction flips to a target class $y^* \neq y$, while maintaining proximity to $\boldsymbol{x}$.

**Class-Conditional Density.** A critical aspect of robust counterfactual generation is the alignment with the underlying data distribution. Let $p_{\text{data}}(\boldsymbol{x})$ denote the marginal data density. However, relying solely on $p_{\text{data}}(\boldsymbol{x})$ is insufficient for CEs, as high-density regions may belong to the original class $y$, leading to trivial or adversarial solutions. Here, we define the *target class-conditional density* as $p(\boldsymbol{x}|y^*)$. This metric quantifies how well an instance $\boldsymbol{x}$ conforms to the manifold of the target class $y^*$. We assume access to a density estimator(discussed in the later section) that approximates this likelihood. Formally, a valid and robust counterfactual $\boldsymbol{x}'$ should satisfy:

$$\boldsymbol{x}' \in \mathcal{X}_{\text{trust}}(y^*) := \{\boldsymbol{x} \in \mathcal{X} \mid p(\boldsymbol{x}|y^*) \geq \tau \cdot p_{\text{ref}}\} \quad (1)$$

where $p_{\text{ref}}$ denotes the reference distribution term, which serves as the background distribution for NCE-based trust-region construction. This definition ensures that the generated counterfactuals are semantically grounded in the distribution of the target class. The threshold $\tau$ explicitly controls the trade-off between robustness and feasibility. A high $\tau$ restricts the search to the varying modes (core) of the class distribution, guaranteeing high robustness and model agreement, but potentially increasing the transport cost $c(\boldsymbol{x}, \boldsymbol{x}')$ or making the counterfactual unreachable due to the overly strict manifold constraints. In practice, we parameterize the threshold $\tau$ via the data–noise sampling ratio, i.e., $N_{\text{noise}}/N_{\text{data}}$ for calibration. We provide a detailed discussion in Appendix C.5.

### 3.2. RCE as Density-Constrained Optimal Transport

We study RCE under model multiplicity (MM). Similar to recent work, such as (Jiang et al., 2024b), we consider a finite set of plausible predictors $\mathcal{M} = (h_j)_{j=1}^m$, where $m$ denotes the number of candidate models, and each $h_j : \mathcal{X} \to \mathcal{Y}$ is a well-trained predictor for the same classification task, but different models may induce different decision boundaries. Accordingly, MM-based robustness asks whether a counterfactual remains valid across this model set, rather than only for one fixed predictor.

From a generative perspective, we reframe the CE problem as an Optimal Transport (OT) task. Let the query instance be a Dirac measure $\mu_0 = \delta_{\boldsymbol{x}}$ and the target be the conditional distribution $\mu_{y^*} = p(\cdot|y^*)$. The goal is to transport mass from $\mu_0$ to $\mu_{y^*}$ with minimal cost. To avoid the "long-tail" fragility discussed earlier, we explicitly enforce that the transported sample must reside within the high-density manifold. We formally define this Density-Guided RCE problem as:

$$\begin{aligned} \boldsymbol{x}_{\text{RCE}}^* = \arg\min_{\boldsymbol{x}'} \quad & c(\boldsymbol{x}, \boldsymbol{x}') \\ \text{s.t.} \quad & \mathbb{E}_{\mathcal{M}}[h(\boldsymbol{x}')] = y^*, \quad \text{(Validity)} \\ & \boldsymbol{x}' \in \mathcal{X}_{\text{trust}}(y^*) \quad \text{(Density)} \end{aligned} \quad (2)$$

Here, $\mathcal{X}_{\text{trust}}(y^*)$ acts as a density barrier. In practice, directly solving Eq. (2) as a static constrained optimization is challenging. Instead, we propose to model the continuous transport trajectory via a Neural ODE. In this formulation, the density constraint is naturally incorporated as a guidance potential, steering the flow dynamics to terminate strictly within the reliable support of the target class.

## 4. Proposed Method

To provide a tractable solution to the density-constrained optimization in Eq. (2), we propose DensityFlow, a coupled framework illustrated in Fig. 2. The framework consists of two primary components jointly optimized to ensure robustness and validity: (i) A NCE-based surrogate ($f_\phi$): we introduce a unified surrogate that simultaneously performs classification and NCE-based density estimation. This surrogate provides differentiable gradients for both target validity and density constraints, steering the trajectory $v_\theta$ away from low-density regions. (ii) A velocity field $v_\theta$ that generates continuous trajectories while minimizing kinetic energy to obtain counterfactual samples at minimal cost. To allow for effective usage in other classifiers, we also design an effective local distillation strategy for alignment.

### 4.1. NCE-based Conditional Density Estimation

To effectively constrain CEs within the trust regions $\mathcal{X}_{\text{trust}}(y^*)$, we need to explicitly estimate the class-conditional density $p(\boldsymbol{x}|y^*)$ across the input space $\mathcal{X} \subseteq \mathbb{R}^d$.

We address this by reformulating density estimation into a Noise Contrastive Estimation (NCE) framework (Gutmann & Hyvärinen, 2012), which learns the density ratio between the data distribution and a known reference noise distribution. Specifically, we design a surrogate classifier $f_\phi : \mathcal{X} \to \mathbb{R}^{K+1}$ that operates on an augmented label space $\{1, \ldots, K, K+1\}$, where the first $K$ indices correspond to semantic classes and the $(K+1)$-th index represents an auxiliary noise class. We construct the training set by mixing the original source data $\mathcal{D}_{\text{src}}$ with noise samples drawn from a parametric distribution $p_{\text{noise}}(\boldsymbol{x})$ (e.g., Uniform or Gaussian). Since $p_{\text{noise}}$ is analytically known, distinguishing real data from noise allows the classifier to implicitly model the underlying data density. The specific impact of the noise distribution is discussed in detail in Sec. 5.5 and Appendix C.5. The surrogate $f_\phi$ is trained to minimize the Cross-Entropy

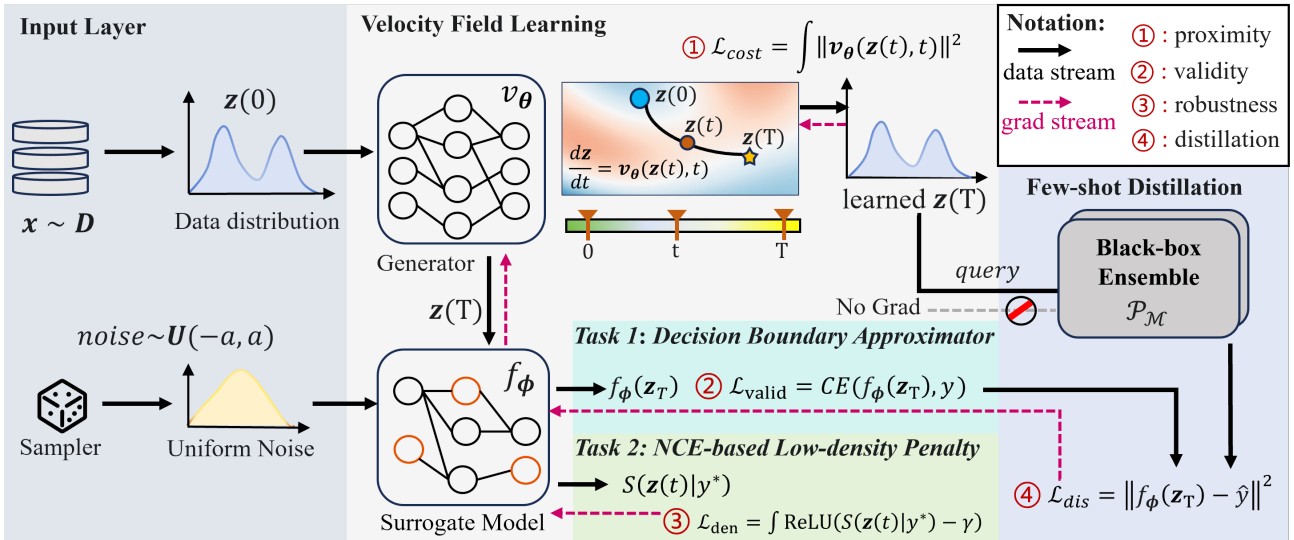

*Figure 2.* Overview of DensityFlow. The framework couples two primary components: (i) A unified surrogate ($f_\phi$) via Noise Contrastive Estimation to model the high-density trust regions and decision boundaries. (ii) A Neural ODE Generator ($v_\theta$) produces a continuous trajectory $z(t)$ that transforms input $x$ into a valid counterfactual. The optimization integrates kinetic regularization for minimal cost, density penalties for robustness, and a local distillation strategy to align with black-box ensembles efficiently.

loss over this $(K+1)$-way discrimination task:

$$
\begin{aligned}
\mathcal{L}_{\text{surrogate}}(\phi) = &- \mathbb{E}_{(\boldsymbol{x},y)\sim \mathcal{D}_{\text{src}}} \left[ \log \frac{e^{z_y(\boldsymbol{x})}}{\sum_{j=1}^{K+1} e^{z_j(\boldsymbol{x})}} \right] \\
&- \mathbb{E}_{\boldsymbol{x}\sim p_{\text{noise}}} \left[ \log \frac{e^{z_{K+1}(\boldsymbol{x})}}{\sum_{j=1}^{K+1} e^{z_j(\boldsymbol{x})}} \right]
\end{aligned}
\tag{3}
$$

$f_\phi$ produces logits $z_k(\boldsymbol{x})$ for $k \in \{1, \ldots, K+1\}$ over the input space $\mathcal{X}$. The model predicts the class $k^*$ corresponding to the maximum logit value. Under the NCE framework, assuming $f^*$ is the optimal minimizer of Eq. (3), we derive the following proposition:

**Proposition 4.1** (Relation to Density Ratio). *For any $\boldsymbol{x} \in \mathcal{X}$ and target class $k \in \{1, \ldots, K\}$, the optimal density score satisfies:*

$$
z_k^*(\boldsymbol{x}) - z_{K+1}^*(\boldsymbol{x}) = \log p(\boldsymbol{x}|k) + Const \tag{4}
$$

*where $z^*$ denotes the optimal logits, and $Const = \log \frac{P(y=k)}{P(y=K+1)} - \log p_{noise}(\boldsymbol{x})$ depends on class priors and the noise density.*

*Proof Sketch.* Since we employ a Uniform Distribution for the noise class, $p_{\text{noise}}(\boldsymbol{x})$ is constant over the input space. Consequently, the logit difference $z_k^*(\boldsymbol{x}) - z_{K+1}^*(\boldsymbol{x})$ becomes linearly proportional to the log-likelihood $\log p(\boldsymbol{x}|k)$. A detailed derivation is provided in Appendix A.1.

**Noise as a trust reference.** The uniform noise also serves as an effective baseline for defining the trust region $\mathcal{X}_{\text{trust}}$.

The ratio of noise to data samples acts as a hyperparameter controlling the strictness of this region: an excessive noise prior leads to an overly conservative boundary, while insufficient noise fails to filter out low-density outliers.

### 4.2. Density-Guided Flow Optimization

We optimize the generator $v_\theta$ using an alternating strategy: in each iteration, we first refine the surrogate $f_\phi$ via Eq. (3), and subsequently update $v_\theta$ to construct optimal counterfactual trajectories. The generation process is driven by three coupled objectives: minimizing transport cost, ensuring validity, and enforcing manifold constraints.

**Cost Estimation via Augmented Dynamics.** To strictly minimize the continuous transport effort, we adopt the state-augmented ODE framework (Lipman et al., 2023). We extend the state space to $\tilde{\boldsymbol{z}}(t) = [\boldsymbol{z}(t), e(t)]^\top$, where $e(t)$ represents the accumulated kinetic energy. The augmented dynamics evolve as:

$$
\frac{d\tilde{\boldsymbol{z}}(t)}{dt} = \begin{bmatrix} v_\theta(\boldsymbol{z}(t), t) \\ \|v_\theta(\boldsymbol{z}(t), t)\|^2 \end{bmatrix}, \quad \tilde{\boldsymbol{z}}(0) = \begin{bmatrix} \boldsymbol{x} \\ 0 \end{bmatrix} \tag{5}
$$

By integrating to $T$, the final state yields the exact transport cost $e(T) = \int_0^T \|v_\theta(\boldsymbol{z}(t), t)\|^2 dt$, enabling direct backpropagation through the ODE solver to minimize $\mathcal{L}_{\text{cost}} \triangleq e(T)$.

**Differentiable Surrogate Guidance.** Since the black-box target is non-differentiable, we leverage the surrogate $f_\phi$ to provide two distinct guidance signals for the generator: (i) Validity proxy: The classification head ensures the generated $\boldsymbol{z}(T)$ flips to the target class $y^*$. (ii) Density critic: The logit

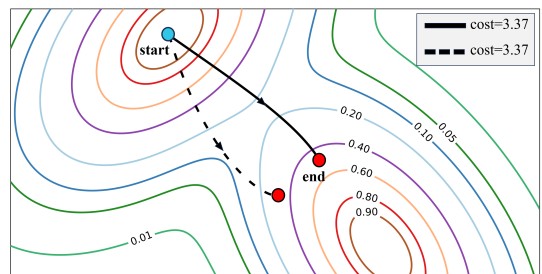

*Figure 3.* **Trajectory comparison on a density surface.** The standard distance-minimization approach (dashed) traverses low-density regions, leading to fragile CEs. In contrast, DensityFlow (solid), guided by the density gradient $\nabla S$, follows the high-probability manifold to ensure robustness.

difference $S(\boldsymbol{z}|y^*) = z_{y^*} - z_{K+1}$ acts as a density critic.

As derived in Proposition 4.1, the gradient of this score serves as an unbiased estimator of the data density gradient: $\nabla_{\boldsymbol{z}} S(\boldsymbol{z}|y^*) = \nabla_{\boldsymbol{z}} \log p(\boldsymbol{z}|y^*)$. As shown in Fig. 3, using this gradient as guidance yields trajectories that avoid low-support regions (dashed) and instead track the high-density manifold (solid).

**Composite Generator Objective.** Formally, to strictly enforce the trust region constraint throughout the generation process, the density penalty can be formulated as a path integral:

$$\mathcal{L}_{\text{den}} = \int_0^T \text{ReLU}(\log \tau - S(\boldsymbol{z}(t)|y^*))dt \qquad (6)$$

However, evaluating the score at every solver step incurs significant computational overhead. In practice, we find that enforcing the constraint only at the endpoint is sufficient to pull the entire trajectory into the valid manifold due to the smoothness of Neural ODEs. Thus, for computational efficiency, we adopt the endpoint penalty in our implementation (see Appendix C.1 for comparison). The final objective for the generator is:

$$\mathcal{L}(\theta) = \underbrace{\mathcal{L}_{\text{CE}}(f_\phi(\boldsymbol{x}'), y^*)}_{\text{Validity}} + \lambda_{\text{cost}} \underbrace{c_{\text{cost}}(T)}_{\text{Proximity}}$$
$$+ \lambda_{\text{den}} \underbrace{\mathbb{E}_{\boldsymbol{x}} \left[\mathcal{L}_{\text{den}}(\boldsymbol{x}')\right]}_{\text{Robustness}} \qquad (7)$$

### 4.3. Query-Efficient Local Distillation

In real-world applications, the target system is often a heterogeneous black-box ensemble $\mathcal{M}$ (Jiang et al., 2024b). Since its architecture differs from our surrogate $f_\phi$, their decision boundaries naturally differ. To generate valid counterfactuals, we must align $f_\phi$ with the ensemble.

A major challenge is that we do not know *where* the decision boundaries differ. A naive approach is to align $f_\phi$ globally

across the entire input space. However, this is computationally intractable. In high-dimensional spaces, we lack prior knowledge of where the boundaries diverge. Without guidance, finding these misalignment regions requires blindly querying the black box across the vast volume of the input space, leading to prohibitive costs. To solve this, we propose a query-efficient strategy that aligns $f_\phi$ strictly within the regions visited by our generator. We dynamically sample a support set $\mathcal{D}_\theta$ from the terminal states of the current flow: $\mathcal{D}_\theta = \{(\boldsymbol{x}, \bar{y}) \mid \boldsymbol{x} \sim \boldsymbol{z}(T)\}$, where $\bar{y}$ is the ensemble consensus. We then fine-tune $f_\phi$ to minimize the local distillation loss:

$$\mathcal{L}_{\text{dis}}(\phi) \triangleq \mathbb{E}_{(\boldsymbol{x},\bar{y})\sim\mathcal{D}_\theta} \left[\|\sigma(z_{y^*}(\boldsymbol{x})) - \bar{y}\|^2\right] \qquad (8)$$

where $\sigma(\cdot)$ denotes the softmax probability for the target class. Eq. (8) ensures $f_\phi$ provides accurate gradients specifically along the generation path.

The key advantage is that density guidance tells us exactly where to align. Without density constraints, a generator might explore useless low-density areas (outliers), forcing us to waste queries aligning boundaries in regions that do not matter. In contrast, our density gradient $\nabla S$ acts as a guide, steering the trajectory strictly towards high-density data manifolds. As a result, we only query the black box in meaningful, high-confidence regions. By avoiding wasted queries on "empty" or low-density spaces, we achieve accurate alignment with significantly lower cost.

## 5. Experiments

To validate the effectiveness of DensityFlow, we designed experiments to answer three key research questions: **RQ1 (Robustness & Efficiency):** Does DensityFlow achieve superior counterfactual robustness while maintaining low query costs? **RQ2 (Density Effectiveness):** Can the proposed density score $S(\boldsymbol{x}|y^*)$ effectively serve as an indicator for identifying trustworthy regions? **RQ3 (Alignment Reliability):** Does local boundary alignment via proxy distillation enable reliable optimization for heterogeneous black-box ensembles? Due to page constraints, additional ablation studies and visualizations are reported in Appendix C.

### 5.1. Experiment Settings

**Datasets.** Our method is evaluated on eight datasets: four synthetic datasets (Moons, Circles, Spirals, and Chessboard) and four tabular real-world datasets (Adult, Compas, HELOC, and Blood), covering diverse domains, including finance, justice, and medicine. We randomly split all datasets into training, validation, and test sets with a 6:2:2 ratio. Prior to training, we apply Z-score standardization to the input features to ensure consistent feature scaling, which simplifies the construction of the noise. The noise samples

are drawn from a uniform distribution bounded within a hypercube $[-C, C]^d$. We set the noise-to-data sampling ratio to $\tau = N_{\text{noise}}/N_{\text{data}} = 0.2$. To ensure the noise fully covers the standardized data support, the bound is defined as $C = \beta \cdot \max_{x \in \mathcal{D}_{\text{train}}} \|x\|_\infty$, where the maximum absolute feature value is scaled by a safety factor $\beta = 1.2$.

**Evaluation Models(Target Ensemble).** To simulate a realistic *Model Multiplicity (MM)* scenario characterized by architectural heterogeneity, we instantiate the target system $\mathcal{M}$ using seven diverse classifiers: KNN (Cover & Hart, 1967), SVM (Cortes & Vapnik, 1995), Random Forest (Breiman, 2001a), MLP, XGBoost (Chen et al., 2016), CatBoost (Prokhorenkova et al., 2018), and TabNet (Arik & Pfister, 2021). Each model represents a distinct inductive bias (e.g., distance-based, tree-based, deep learning). We fine-tune their hyperparameters on the validation set to ensure strong individual performance (see Table 1 for accuracy statistics). This diversity ensures that $\mathcal{M}$ provides a rigorous testbed for consensus-based explanation. Detailed settings and results are presented in Table 5 & 6 in the Appendix.

*Table 1.* Dataset statistics and average model accuracy.

| Dataset | Rows | Features | Avg. Accuracy |
|---|---|---|---|
| Moons | 3000 | 2 | 0.9998 (.000) |
| Circles | 3000 | 2 | 0.9989 (.000) |
| Spirals | 3000 | 2 | 0.9859 (.004) |
| Chessboard | 3000 | 2 | 0.9992 (.001) |
| Adult | 48842 | 12 | 0.8579 (.001) |
| Compas | 7214 | 9 | 0.6753 (.003) |
| HELOC | 10459 | 23 | 0.7106 (.002) |
| Blood | 748 | 4 | 0.7994 (.004) |

**Baselines.** We compare our DensityFlow against four state-of-the-art robust recourse methods : Product_mip (Leofante et al., 2023), CeFlow (Duong et al., 2023), Argument (Jiang et al., 2024a), and BetaRCE (Stępka et al., 2025). Please note that *Product_mip* and *CeFlow* are designed for white-box; therefore, we use a well-tuned MLP instead of the white-box model and then evaluate it on $\mathcal{M}$. *BetaRCE* is designed for handling mild changes in a black-box model; here, we use $\mathcal{M}$ instead of the so-called 'admissible model space'. The *Argument* is closest to our setting, as it considers recourse under MM and supports a heterogeneous black-box scenario. However, this method differs from ours in that it does not generate counterfactuals. Instead, it assumes the availability of one counterfactual per model and focuses on post-hoc selection via an argumentation framework to ensure consistency and validity. In contrast, DensityFlow is a generation framework that directly optimizes multi-model validity during generation. For the detailed parameter settings of DensityFlow and the baseline methods, please refer to Appendix B.4.

**Parameters.** Unless otherwise specified, we train for 800 epochs with a batch size of 64 using AdamW, with learning rates $\eta_g = 10^{-3}$ (generator $v_\theta$) and $\eta_\phi = 10^{-4}$ (surrogate $f_\phi$). The objective weights were selected via grid search from the candidate sets: $\lambda_{\text{cost}} \in \{0.2, 0.4, 0.6\}$, and $\lambda_{\text{den}} \in \{0.0, 0.1, 0.3\}$. Neural ODE integration uses the adaptive `dopri5` solver on $t \in [0, 1]$ with 100 evaluation time points and tolerances $(\texttt{atol}, \texttt{rtol}) = (10^{-3}, 10^{-3})$ for training and $(10^{-4}, 10^{-4})$ for testing. This adaptive step-size mechanism automatically adjusts to local error tolerances, ensuring finer sampling in regions with complex density gradients and preventing the trajectories from deviating from the density guidance. All experiments were conducted on a server with NVIDIA GeForce RTX 4090 GPUs, AMD EPYC 7513, and 512 GB RAM. Additional details are provided in Appendix B.3.

**Metrics.** Similar to previous studies (Jiang et al., 2024a; Noorani et al., 2025), we focus on two metrics: (i) Cost: the $\ell_2$ distance between the counterfactual and the corresponding original instance; (ii) Validity: the proportion of generated counterfactuals that are successfully classified as the target class by the ensemble models. All results are reported as averages over 5 independent runs (seeds 0∼4).

### 5.2. RQ1: Performance under Model Multiplicity

First, we compare the CEs learned by DensityFlow with the baselines on eight synthetic and real-world datasets. Table 2 presents the validity and costs evaluated across the black-box ensemble $\mathcal{M}$. As shown in the results, DensityFlow consistently achieves high validity while maintaining competitive costs. Specifically, on real-world tabular datasets such as Adult and Compas, our method surpasses the strongest baseline by a significant margin in validity(e.g., 0.901 vs. 0.752 on Adult), demonstrating its effectiveness in navigating complex decision boundaries. On synthetic datasets like Moons and Circles, DensityFlow achieves near-perfect validity ($> 99\%$). This increase in robustness does not sacrifice proximity. On Adult, DensityFlow achieves a cost of 1.597, compared to 3.959 for CeFlow and 1.916 for Argument, effectively balancing the validity-cost trade-off.

Product_mip relies on access to internal model parameters and relational verification, which restricts its applicability in strictly black-box scenarios. BetaRCE is designed for a specific target model and can be less straightforward to apply to heterogeneous black-box ensembles. While CeFlow promotes manifold adherence, it lacks an explicit mechanism to bypass low-density regions, frequently generating CEs that reside within 'fuzzy' decision boundaries. Argument performs post-hoc selection over a candidate set; when candidates are not support-aware, the resulting trade-off between robustness and proximity can be suboptimal. In contrast, DensityFlow embeds robustness optimization directly into the Neural ODE dynamics. By leveraging $S(\boldsymbol{x}|y^*)$ guid-

*Table 2.* Performance comparison of all methods on the black-box ensemble $\mathcal{M}$. **Cost**($\downarrow$) is the $\ell_2$ distance between $\boldsymbol{x}$ and $\boldsymbol{x}'$, lower the better; **Validity**($\uparrow$) is the average proportion of valid CEs in $\mathcal{M}$, higher the better.

| Dataset | Product_mip | | CeFlow | | BetaRCE | | Argument | | DensityFlow | |
|---|---|---|---|---|---|---|---|---|---|---|
| | Cost | Validity | Cost | Validity | Cost | Validity | Cost | Validity | Cost | Validity |
| Moons | 1.214±.463 | 0.492±.005 | 1.194±.252 | 0.991±.004 | 0.992±.082 | 0.789±.004 | 0.890±.109 | 0.951±.042 | **0.867**±**.101** | **0.997**±**.002** |
| Circles | **0.512**±**.117** | 0.637±.014 | 1.195±.087 | 0.845±.027 | 0.605±.162 | 0.762±.031 | 0.699±.097 | 0.991±.001 | 0.683±.008 | **0.994**±**.001** |
| Spirals | 1.262±.510 | 0.482±.003 | 0.895±.104 | 0.877±.074 | **0.220**±**.084** | 0.631±.060 | 0.356±.034 | 0.943±.067 | 0.487±.139 | **0.972**±**.005** |
| Chessboard | 0.940±.629 | 0.332±.022 | 0.804±.071 | 0.575±.008 | 0.729±**.408** | 0.622±.235 | 1.114±.028 | 0.959±.013 | 1.088±.152 | **0.964**±**.011** |
| Adult | 3.725±.006 | 0.687±.001 | 3.959±.463 | 0.691±.064 | 2.610±.101 | 0.752±.010 | 1.916±.025 | 0.659±.011 | **1.597**±**.194** | **0.901**±**.052** |
| Compas | 1.631±.017 | 0.479±.005 | **0.995**±**.282** | 0.385±.058 | 1.410±.005 | 0.592±.003 | 1.542±.001 | 0.610±.006 | 1.176±.017 | **0.729**±**.067** |
| HELOC | 3.148±.709 | 0.444±.044 | 2.707±.085 | 0.542±.023 | **1.640**±**.122** | **0.765**±**.016** | 3.091±.026 | 0.662±.018 | 1.812±.599 | 0.757±.035 |
| Blood | **1.454**±**.038** | 0.378±.018 | 1.556±.142 | 0.414±.060 | 1.573±.045 | 0.285±.192 | 1.548±.063 | 0.509±.007 | 1.527±.401 | **0.662**±**.046** |

ance, trajectories autonomously avoid low-density regions. This approach enhances robustness on complex manifolds without a significant increase in modification costs. In summary, DensityFlow achieves higher robust validity across the ensemble on most datasets while keeping costs competitive with the strongest baselines.

### 5.3. RQ2: Correlation between Density and Robustness

To empirically validate Eq. (4), we quantify the ensemble uncertainty using Mutual Information (MI), a standard measure for epistemic uncertainty (Smith & Gal, 2018). MI is defined as the difference between the entropy of the mean prediction and the mean entropy of individual predictions:

$$\text{Uncertainty}(\boldsymbol{x}) \triangleq \mathbb{H}[\bar{p}(y|\boldsymbol{x})] - \frac{1}{M}\sum_{m=1}^{M}\mathbb{H}[p_m(y|\boldsymbol{x})] \quad (9)$$

where $\mathbb{H}[\cdot]$ denotes Shannon entropy and $\bar{p}$ is the ensemble mean. A high MI value indicates significant uncertainty in the ensemble's findings and low reliability. We visualize the correlation between the density score $S(\boldsymbol{x}|y^*)$ and MI. Fig. 4 illustrates the relationship between density score

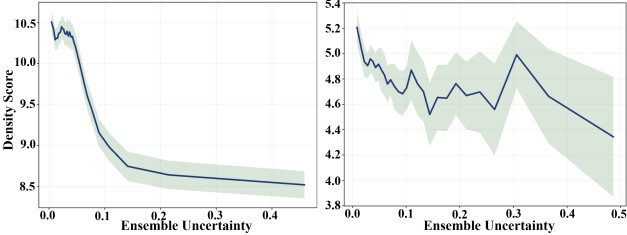

*Figure 4.* Correlation between density score $S(\boldsymbol{x}|y^*)$ and ensemble uncertainty (measured by MI) on Adult (left) and HELOC (right) datasets. The bands represent 95% confidence intervals.

and model uncertainty. On the Adult dataset, we observe a clear relationship: as the density score decreases (towards low-density tails), ensemble disagreement rises sharply, confirming that $S(\boldsymbol{x}|y^*)$ effectively tracks the safe consensus regions. For the HELOC dataset, the curves exhibit local oscillations, but the overall trend remains consistent, which is

normal given the inherent complexity of the dataset (such as noise and outliers). Crucially, samples with the high density scores consistently align with minimal model disagreement. This demonstrates that, even in noisy real-world scenarios where classifiers struggle to generalize, our density scores serve as a reliable indicator, guiding CEs to the trust region.

### 5.4. RQ3: Query Efficiency for Black-Box Ensemble

We evaluate the query efficiency of DensityFlow by measuring the total number of black-box ensemble queries required during the optimization or training phase. We compare our method with black-box applicable baselines, Argument and BetaRCE. The query count is strictly incremented once for each invocation of the black-box model during training or search, excluding inference-time evaluations.

As shown in Fig. 6 (left), DensityFlow requires fewer queries to obtain valid CEs. Baselines typically rely on iterative queries to search for valid candidates or to verify robustness, whereas DensityFlow confines black-box interaction to a focused local surrogate distillation phase. Fig. 6 (right) illustrates that the validity of CEs saturates rapidly with respect to the number of distillation queries. This behavior indicates that the local decision boundary can be effectively approximated with a small query budget. We emphasize, however, that due to differences in mechanism and scenario across methods, these results should not be interpreted as an overall judgment of the superiority of one paradigm over another. In summary, the designed distillation strategy allows DensityFlow to operate effectively in query-restricted black-box scenarios.

*Table 3.* Ablation Study. *w/o Density* denotes the variant with the NCE module removed, and *w/o Distill* indicates the exclusion of the distillation alignment module.

| Dataset | Densityflow | w/o Density | w/o Distill |
|---|---|---|---|
| **Adult** | $0.901 \pm .052$ | $0.815 \pm .132$ | $0.767 \pm .044$ |
| **Blood** | $0.662 \pm .046$ | $0.495 \pm .033$ | $0.531 \pm .004$ |
| **Compas** | $0.729 \pm .067$ | $0.642 \pm .052$ | $0.698 \pm .054$ |
| **HELOC** | $0.757 \pm .036$ | $0.718 \pm .095$ | $0.734 \pm .038$ |

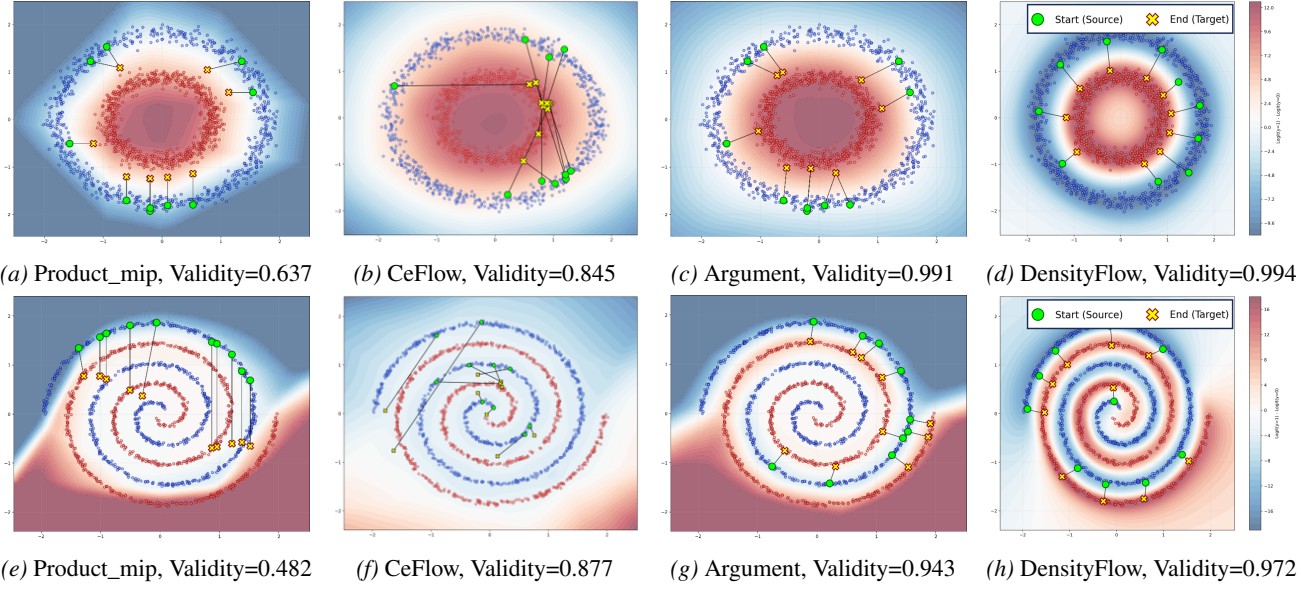

*(a)* Product_mip, Validity=0.637    *(b)* CeFlow, Validity=0.845    *(c)* Argument, Validity=0.991    *(d)* DensityFlow, Validity=0.994

*(e)* Product_mip, Validity=0.482    *(f)* CeFlow, Validity=0.877    *(g)* Argument, Validity=0.943    *(h)* DensityFlow, Validity=0.972

*Figure 5.* Visualization results on Circles and Spirals. For more visualizations, please refer to Appendix D.

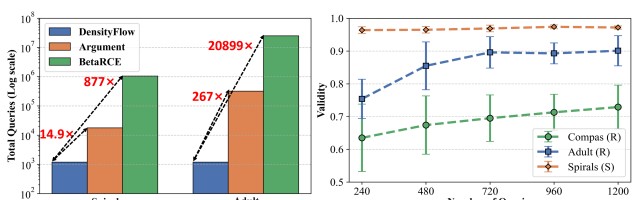

*Figure 6.* **Left:** Comparison of total query costs with state-of-the-art black-box methods, Argument and BetaRCE, on Spirals and Adult dataset (log scale). **Right:** Impact of the internal distillation budget on validity. The curve shows that our method achieves high performance even with a restricted query budget.

*Figure 7.* Validity (solid) and Cost (dashed) for varying $\tau$. In the low-ratio regime, sparse noise leads to performance degradation. However, performance rapidly converges to a stable plateau as $\tau$ increases, indicating insensitivity to large noise.

## 5.5. Ablation Study

Table 3 shows removing the density term consistently reduces validity across all datasets, highlighting the importance of density guidance in discouraging low-density regions for RCEs. Removing distillation further degrades performance, with a particularly severe collapse on Blood. Notably, even without the distillation alignment, DensityFlow still achieves the best results on three of the four real-world datasets. We also explore replacing the NCE with off-the-shelf density proxies (e.g., kNN or LOF) under fixed settings and observe improvements over removing the density signal in Appendix C.2.

**Impact of the noise ratio $\tau$.** In this part, we analyze the noise ratio $\tau$. As shown in Fig. 7, varying $\tau$ reveals a trade-off between validity and cost. At low ratios, both validity and cost drop sharply (e.g., Compas cost falls to $\approx 0.75$). This happens because sparse noise cannot clearly define the data boundary. Without this boundary, the generator takes shortcuts through low-density regions, creating adversar-

ial examples instead of valid counterfactuals. Increasing $\tau$ tightens the trust region, forcing the generator to traverse further into the high-density manifold. This naturally improves validity but incurs a higher transport cost. While performance stabilizes after a certain point in our experiments, the optimal $\tau$ is not a universal constant; it depends on the intrinsic sparsity of the dataset and can be tuned to balance the desired robustness against the acceptable cost. We further analyze other hyperparameters in Appendix C.4.

## 5.6. Case Visualization on Synthetic Datasets

Finally, we compare the counterfactual generated by DensityFlow with the baselines on two synthetic datasets and provide a visual analysis. As illustrated in Fig. 5, DensityFlow generates smooth trajectories that adhere to the data manifold. NCE explicitly models the gaps between manifolds and sparse regions in the feature space, which appear

as white or light-colored areas.

## 6. Conclusion

We presented DensityFlow, a framework that redefines robust counterfactual generation as a density-guided optimal transport problem. We couple Neural ODEs and NCE with joint optimization to guide trajectories away from low-density regions, navigating the trade-off between robustness and cost. To address the efficiency challenges in black-box settings, we introduced a localized alignment strategy that distills the target models within the high-density region, achieving low query complexity. Extensive experiments show that DensityFlow excels in both effectiveness and robustness while maintaining low query complexity.

**Limitations and future work.** First, scaling the density-score component to very high-dimensional data is non-trivial. A feasible strategy to mitigate this is employing feature selection (e.g., via LassoNet (Lemhadri et al., 2021)) to isolate the most predictive features before generating CEs. However, this approach may lead to causal coupling: discarded 'non-predictive' features might be physically or causally coupled to the features modified by the counterfactual. For example, drastically altering a predictive feature like 'pH' without properly adjusting a filtered, non-predictive feature like 'conductivity' can yield out-of-distribution CEs. Furthermore, because our framework relies on learning class-conditional support, its density guidance can be weakened by extreme class imbalance or noisy labels. Finally, unlike discovering rare but interesting "edge cases" (e.g., CFKD (Bender et al., 2023)), this paper focuses on the robustness of the interpretation. Future work will explore implementing density estimation at the embedding level (e.g., leveraging pre-trained representations) for complex modalities and establishing reliability criteria for the local proxy under severe distribution shifts.

## Impact Statement

Approaches that steer CEs away from low-density regions of the data can, in principle, make recommendations more consistent and less likely to rely on extreme or unlikely feature configurations. At the same time, data support is not a perfect stand-in for legitimacy: some meaningful cases are genuinely rare, and some subpopulations may be sparsely represented in historical data. For these reasons, we view distribution-aware guidance as a complement to, rather than a replacement for, application-specific constraints and careful evaluation, including reporting outcomes for tail cases and relevant subgroups when counterfactuals are used in high-stakes settings.

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

# A. Omitted Proof and Additional Derivations

## A.1. Detailed Proofs of Density Scoring (Proposition 4.1)

This section provides the rigorous derivation connecting our surrogate classification objective to the target class-conditional density.

**Restatement of Proposition 4.1.** *Let $f^*$ be the optimal classifier minimizing the population risk of Eq. (3). Assuming the class priors are $\pi_k = P(y = k)$ for semantic classes and $\pi_{K+1}$ for the noise class. For any $\boldsymbol{x} \in \mathcal{X}$ and target class $y^* \in \{1, \ldots, K\}$, the optimal score difference satisfies:*

$$\Delta z^*(\boldsymbol{x}) \triangleq z_{y^*}^*(\boldsymbol{x}) - z_{K+1}^*(\boldsymbol{x}) = \log p(\boldsymbol{x}|y^*) + C \tag{10}$$

*where $C$ is a constant determined by the priors and the noise density.*

### A.1.1. PROOF OF PROPOSITION 4.1

We first establish the form of the optimal discriminator output.

**Lemma A.1** (Optimal Logit Relationship). *Consider a classification task where samples are drawn from a mixture distribution with priors $\pi_k$ and class-conditional densities $p_k(\boldsymbol{x})$. Minimizing the cross-entropy loss forces the optimal logits $z_k^*(\boldsymbol{x})$ to satisfy the following relation for any pair of classes $i, j$:*

$$z_i^*(\boldsymbol{x}) - z_j^*(\boldsymbol{x}) = \log(\pi_i p_i(\boldsymbol{x})) - \log(\pi_j p_j(\boldsymbol{x})) \tag{11}$$

*Proof of Lemma A.1.* The population risk for cross-entropy is equivalent to minimizing the Kullback-Leibler (KL) divergence between the true posterior $P(Y = k|\boldsymbol{x})$ and the predicted probability $D_k(\boldsymbol{x}) = \text{softmax}_k(\boldsymbol{z}(\boldsymbol{x}))$. The true posterior is given by Bayes' rule:

$$P(Y = k|\boldsymbol{x}) = \frac{\pi_k p_k(\boldsymbol{x})}{\sum_m \pi_m p_m(\boldsymbol{x})} \tag{12}$$

The optimal classifier $f^*$ matches these probabilities, i.e., $D_k^*(\boldsymbol{x}) = P(Y = k|\boldsymbol{x})$. Since $D_k^*(\boldsymbol{x}) \propto e^{z_k^*}$, the ratio of probabilities for any two classes $i, j$ is:

$$\frac{D_i^*(\boldsymbol{x})}{D_j^*(\boldsymbol{x})} = e^{z_i^* - z_j^*} \tag{13}$$

Equating this to the ratio of true posteriors:

$$e^{z_i^* - z_j^*} = \frac{P(Y = i|\boldsymbol{x})}{P(Y = j|\boldsymbol{x})} = \frac{\pi_i p_i(\boldsymbol{x})}{\pi_j p_j(\boldsymbol{x})} \tag{14}$$

Taking the natural logarithm of both sides yields the lemma. □

**Derivation of the Density Score.** Applying Lemma A.1 with class $i = y^*$ (target) and class $j = K+1$ (noise):

$$\begin{aligned}
z_{y^*}^*(\boldsymbol{x}) - z_{K+1}^*(\boldsymbol{x}) &= \log(\pi_{y^*} p_{y^*}(\boldsymbol{x})) - \log(\pi_{K+1} p_{K+1}(\boldsymbol{x})) \\
&= \log p_{y^*}(\boldsymbol{x}) - \log p_{K+1}(\boldsymbol{x}) + \log \frac{\pi_{y^*}}{\pi_{K+1}}
\end{aligned} \tag{15}$$

Substituting the definitions $p_{y^*}(\boldsymbol{x}) = p(\boldsymbol{x}|y^*)$ and $p_{K+1}(\boldsymbol{x}) = p_{\text{noise}}(\boldsymbol{x})$:

$$z_{y^*}^*(\boldsymbol{x}) - z_{K+1}^*(\boldsymbol{x}) = \log p(\boldsymbol{x}|y^*) \underbrace{- \log p_{\text{noise}}(\boldsymbol{x}) + \log \frac{\pi_{y^*}}{\pi_{K+1}}}_{Const} \tag{16}$$

Since we employ a Uniform Noise distribution, $p_{\text{noise}}(\boldsymbol{x})$ is constant over the domain $\mathcal{X}$. Consequently, the entire underbraced term becomes a constant. This confirms that the logit difference is strictly linear with the target log-density. □

## A.2. Robustness v.s Data Density

Our framework builds on the low-density separation principle (Chapelle & Zien, 2005), which implies that classifier disagreement (epistemic uncertainty) is concentrated in regions with low probability mass (Lakshminarayanan et al., 2017). We formalize this relationship as follows:

**Assumption A.2** (Density-Variance Correlation). Let $\sigma^2_{\mathcal{M}}(\boldsymbol{x})$ denote the predictive variance across a set of plausible models $\mathcal{M}$. For a target class $y^*$, we assume an inverse correlation between class-conditional density and model disagreement:

$$p(\boldsymbol{x}_1|y^*) > p(\boldsymbol{x}_2|y^*) \implies \sigma^2_{\mathcal{M}}(\boldsymbol{x}_1) < \sigma^2_{\mathcal{M}}(\boldsymbol{x}_2) \tag{17}$$

Under this assumption, constraining generated samples to the high-density manifold $\mathcal{X}_{\text{trust}}(y^*; \tau)$ inherently minimizes epistemic uncertainty, thereby theoretically bounding the robust error.

We provide a theoretical justification for using density maximization to improve the robustness of CEs against model multiplicity. Let $\mathcal{H} \sim \mathcal{P}_{\mathcal{M}}$ denote a random classifier drawn from the distribution of plausible models. For a generated counterfactual $\boldsymbol{x}'$, we define its *robustness* as the probability that a random model predicts the target class $y^*$:

$$\text{Rob}(\boldsymbol{x}') \triangleq \Pr_{\mathcal{H} \sim \mathcal{P}_{\mathcal{M}}} [\mathcal{H}(\boldsymbol{x}') = y^*] \tag{18}$$

Let $s_{\mathcal{H}}(\boldsymbol{x}')$ be the model's confidence score for $y^*$ (e.g., softmax output), and $t$ be the decision threshold. We define the mean validity $\mu(\boldsymbol{x}') \triangleq \mathbb{E}_{\mathcal{H}}[s_{\mathcal{H}}(\boldsymbol{x}')]$ and the model disagreement $\sigma^2(\boldsymbol{x}') \triangleq \text{Var}_{\mathcal{H}}[s_{\mathcal{H}}(\boldsymbol{x}')]$.

We aim to maximize $\text{Rob}(\boldsymbol{x}') = \Pr[s_{\mathcal{H}}(\boldsymbol{x}') > t]$. Assuming the counterfactual is valid on average (i.e., $\mu(\boldsymbol{x}') > t$), we apply Cantelli's inequality (one-sided Chebyshev's inequality):

$$\Pr(s_{\mathcal{H}}(\boldsymbol{x}') \leq t) \leq \frac{\sigma^2(\boldsymbol{x}')}{\sigma^2(\boldsymbol{x}') + (\mu(\boldsymbol{x}') - t)^2} \tag{19}$$

The lower bound for robustness is therefore:

$$\text{Rob}(\boldsymbol{x}') \geq 1 - \underbrace{\frac{\sigma^2(\boldsymbol{x}')}{\sigma^2(\boldsymbol{x}') + (\mu(\boldsymbol{x}') - t)^2}}_{\text{Failure Risk Term}} \tag{20}$$

Eq. (20) indicates that to tighten the robustness lower bound, one must minimize the failure risk. Assuming $\mu(\boldsymbol{x}')$ remains stable, this is primarily achieved by reducing the predictive variance $\sigma^2(\boldsymbol{x}')$. By Assumption A.2, $\sigma^2(\boldsymbol{x}')$ is inversely correlated with the class-conditional density $p(\boldsymbol{x}'|y^*)$. Since maximizing our derived score difference $z_{y^*} - z_{K+1}$ is equivalent to maximizing this density (as proven in Appendix A.1), our method implicitly suppresses predictive variance, thereby theoretically enhancing the robustness guarantee.

# B. Implementation Details

## B.1. Dataset Description

**Synthetic datasets.** We utilize four synthetic 2D datasets to evaluate the performance of our framework, particularly for visualizing decision boundaries and counterfactual paths. Each dataset consists of $n = 3,000$ instances with $d = 2$ numerical features, which are normalized using `StandardScaler` to ensure consistent feature scaling.

- **Moons**: This dataset is generated using the `make_moons` function from *scikit-learn*. It forms two interleaving half-circle shapes with a Gaussian noise level of 0.05, providing a classic non-linearly separable classification task.

- **Spirals**: This dataset comprises two intertwined spirals with 2.5 rotations. It is constructed using polar coordinates where the radius is proportional to the angle, with added Gaussian noise to create overlapping regions.

- **Circles**: This dataset consists of two concentric circles generated via trigonometric transformations. The outer circle has a radius of 1.0, while the inner circle is scaled by a factor of 0.5, with a noise level of 0.05.

- **Chessboard**: This dataset represents a $3 \times 3$ grid of clusters where labels are assigned based on the parity of the grid indices. This configuration creates a highly fragmented decision boundary, testing the model's ability to navigate discrete cluster-based regions.

The Python implementations for generating these synthetic datasets are illustrated in Figure 8.

```python
# 1. Intertwined Spirals Generation
def make_intertwined_spirals(n_samples=3000, noise=0.1, n_rotations=2.5):
    n_per_class = n_samples // 2
    theta = np.sqrt(np.random.rand(n_per_class)) * n_rotations * 2 * np.pi
    r = theta / (n_rotations * 2 * np.pi) * 5
    X0 = np.column_stack([r * np.cos(theta), r * np.sin(theta)])
    X1 = np.column_stack([r * np.cos(theta + np.pi), r * np.sin(theta + np.pi)])
    return np.vstack([X0, X1]) + noise * np.random.randn(n_samples, 2)

# 2. Concentric Circles Generation
def make_concentric_circles(n_samples=3000, noise=0.05, factor=0.5):
    angles = 2 * np.pi * np.random.rand(n_samples // 2)
    X_outer = np.column_stack([np.cos(angles), np.sin(angles)])
    X_inner = factor * np.column_stack([np.cos(angles), np.sin(angles)])
    return np.vstack([X_outer, X_inner]) + noise * np.random.randn(n_samples, 2)

# 3. Chessboard Grid Generation
def make_Chessboard(n_samples=3000, n_clusters=3, noise=0.1):
    X, y = [], []
    for i in range(n_clusters):
        for j in range(n_clusters):
            center = [(i + 0.5) / n_clusters * 6 - 3, (j + 0.5) / n_clusters * 6 - 3]
            X.append(center + noise * np.random.randn(n_samples // 9, 2))
            y.append(np.full(n_samples // 9, (i + j) % 2))
    return np.vstack(X), np.hstack(y)
```

*Figure 8.* Implementation details for synthetic dataset generation logic.

**Real-world datasets.** In addition to synthetic data, we evaluate our method on four widely used real-world tabular datasets: *Adult*, *Blood*, *Compas*, and *HELOC*. All datasets are used for classification tasks and consist of numerical, categorical, or mixed-type features.

*Table 4.* Statistics of real-world datasets.

| Dataset | #Rows | #Features | Numerical | Categorical | Feature type |
|---------|-------|-----------|-----------|-------------|--------------|
| Adult | 48,842 | 12 | 4 | 8 | Mixed |
| Blood | 748 | 4 | 4 | 0 | Numerical |
| Compas | 7,214 | 9 | 0 | 9 | Categorical |
| HELOC | 10,459 | 23 | 23 | 0 | Numerical |

In Table 4, #Rows denotes the total number of samples, #Features represents the total number of features, and the columns *Numerical* and *Categorical* indicate the number of numerical and categorical features, respectively. The *Feature type* column summarizes the data type: numerical, categorical, or mixed. Below is a detailed introduction to each dataset:

- **Adult** [1]: This is a binary classification task to predict whether an individual earns more than $50,000 per year based on features such as occupation, marital status, and education.

- **Blood** [2]: This dataset supports a binary classification task to forecast whether a blood donor will make another donation, based on their historical donation records.

---

[1]https://archive.ics.uci.edu/ml/datasets/adult
[2]https://archive.ics.uci.edu/ml/datasets/blood+transfusion+service+center

- **Compas** [3]: A binary classification dataset designed to predict if a defendant will reoffend within two years, using demographic and judicial information.

- **HELOC** [4]: A dataset for binary classification that evaluates credit risk by analyzing applicants' financial and credit-related information.

**Data Preparation** We split each dataset into numerical and categorical features. Numerical features are imputed with the median value computed on the training split and then standardized feature-wise using z-score normalization. For categorical variables, instead of one-hot encoding, we adopt the continuous representation used in TabRep (Si et al., 2025): a category $k_i$ of a $K$-way feature is encoded as

$$\mathbf{E}(k_i) = \left[ \cos\left(\frac{2\pi k_i}{K}\right), \ \sin\left(\frac{2\pi k_i}{K}\right) \right].$$

This yields a low-dimensional continuous representation for mixed tabular inputs. During inference, numerical features are mapped back to the original scale, and each categorical feature is decoded to the nearest valid category on the circle.

### B.2. Evaluation Models

We trained seven classical and strong classifiers, including kNN (Cover & Hart, 1967), SVM (Cortes & Vapnik, 1995), Random Forest (Breiman, 2001a), MLP, XGBoost (Chen et al., 2016), CatBoost (Prokhorenkova et al., 2018) and TabNet (Arik & Pfister, 2021), with hyperparameters tuned on the validation set. The search ranges for each model are summarized in Table 5, and all models used 3-fold cross-validation for parameter selection. These classifiers collectively constitute the target model ensemble $\mathcal{M}$. The complete classification results are presented in Table 6, these classifiers exhibited almost identical classification performance.

*Table 5.* Hyperparameter search ranges for evaluation models.

| Model | Parameter | Search Range |
|---|---|---|
| kNN | $n\_neighbors$ | [3, 5, 7, 9] |
| | $weights$ | ['uniform', 'distance'] |
| MLP | $hidden\_layer\_sizes$ | [(64,64), (128,64), (128,64,64), (128,64,64,128)] |
| | $alpha$ | $[10^{-4}, 10^{-3}, 10^{-2}]$ |
| SVM | $C$ | [1, 10, 100] |
| | $gamma$ | [0.1, 1, 10, 100] |
| | $Kernel$ | RBF |
| Random Forest | $n\_estimators$ | [100, 300] |
| | $max\_depth$ | [3, 5, 10] |
| | $min\_samples\_split$ | [2, 5] |
| XGBoost | $n\_estimators$ | [100, 300] |
| | $max\_depth$ | [3, 5, 7] |
| | $learning\_rate$ | [0.03, 0.1] |
| CatBoost | $depth$ | [4, 6, 8] |
| | $l2\_leaf\_reg$ | [3, 5, 7] |
| | $learning\_rate$ | [0.03, 0.1] |
| TabNet | $mask\_type$ | 'sparsemax' |
| | $max\_epochs$ | [200,500] |
| | $learning\_rate$ | [0.005, 0.03] |

---

[3]https://github.com/propublica/compas-analysis.git

[4]https://community.fico.com/s/explainable-machine-learning-challenge

*Table 6.* Classification accuracy on different models in five different runs(mean $\pm$ std).

| Dataset | kNN | MLP | SVM | RF | XGB | CAT | TabNet |
|---|---|---|---|---|---|---|---|
| Moons | $1.0000 \pm 0.0000$ | $1.0000 \pm 0.0000$ | $1.0000 \pm 0.0000$ | $1.0000 \pm 0.0000$ | $1.0000 \pm 0.0000$ | $1.0000 \pm 0.0000$ | $0.9989 \pm 0.0016$ |
| Circles | $1.0000 \pm 0.0000$ | $0.9989 \pm 0.0016$ | $1.0000 \pm 0.0000$ | $0.9967 \pm 0.0000$ | $0.9967 \pm 0.0000$ | $1.0000 \pm 0.0000$ | $1.0000 \pm 0.0000$ |
| Spirals | $1.0000 \pm 0.0000$ | $0.9633 \pm 0.0027$ | $1.0000 \pm 0.0000$ | $0.9983 \pm 0.0000$ | $0.9967 \pm 0.0000$ | $1.0000 \pm 0.0000$ | $0.9428 \pm 0.0225$ |
| Chessboard | $1.0000 \pm 0.0000$ | $0.9967 \pm 0.0047$ | $1.0000 \pm 0.0000$ | $1.0000 \pm 0.0000$ | $0.9983 \pm 0.0000$ | $0.9994 \pm 0.0000$ | $1.0000 \pm 0.0000$ |
| Adult | $0.8357 \pm 0.0000$ | $0.8553 \pm 0.0007$ | $0.8499 \pm 0.0002$ | $0.8650 \pm 0.0000$ | $0.8753 \pm 0.0000$ | $0.8755 \pm 0.0010$ | $0.8485 \pm 0.0016$ |
| Compas | $0.6500 \pm 0.0000$ | $0.6766 \pm 0.0045$ | $0.6798 \pm 0.0000$ | $0.6824 \pm 0.0027$ | $0.6826 \pm 0.0000$ | $0.6826 \pm 0.0010$ | $0.6784 \pm 0.0086$ |
| HELOC | $0.6955 \pm 0.0000$ | $0.7098 \pm 0.0085$ | $0.7094 \pm 0.0000$ | $0.7177 \pm 0.0006$ | $0.7132 \pm 0.0000$ | $0.7200 \pm 0.0021$ | $0.7084 \pm 0.0014$ |
| Blood | $0.8200 \pm 0.0000$ | $0.7622 \pm 0.0031$ | $0.7933 \pm 0.0000$ | $0.8222 \pm 0.0063$ | $0.8133 \pm 0.0000$ | $0.8067 \pm 0.0109$ | $0.7778 \pm 0.0083$ |

## B.3. Parameter Setting of DensityFlow

Unless otherwise specified, we train DensityFlow with 800 epochs and a batch size of 64. We optimize the generator $v_\theta$ and the surrogate $f_\phi$ using AdamW with weight decay $10^{-4}$, using learning rates $\eta_g = 10^{-3}$ and $\eta_\phi = 10^{-4}$, respectively. We apply a cosine annealing schedule with $\eta_{\min} = 10^{-6}$ and a linear warm-up for the first 20 epochs, and maintain an exponential moving average (EMA) of model parameters with decay 0.999.

For the Neural ODE, we integrate over $t \in [0, 1]$ using an adaptive Dormand–Prince solver (`dopri5`) with tolerances $(\texttt{rtol}, \texttt{atol}) = (10^{-3}, 10^{-3})$ during training and $(10^{-4}, 10^{-4})$ at test time. The noise distribution is defined over a bounded domain $[-C, C]^d$, which also serves as the search space for the generator. The bound $C$ is set adaptively to cover the support of the training data: $C = \max\{2.0, \ \beta \cdot \max_{x \in \mathcal{D}_{\text{train}}} \|x\|_\infty\}$, here we set $\beta = 1.2$, further discussion on $C$ and $\beta$ are presented in Appendix C.5.

For NCE-based density learning, we use a $(K+1)$-class discriminator (semantic classes plus one noise class) and sample uniform noise with ratio $r = 0.2$; the noise classification term is weighted by $\lambda_{\text{noise}} = 1.0$. The objective weights were selected via grid search from the candidate sets: $\lambda_{\text{cost}} \in \{0.2, 0.4, 0.6\}$, and $\lambda_{\text{den}} \in \{0.0, 0.1, 0.3\}$, where the density term is applied as $\text{ReLU}(log\tau - S(\cdot|y^*))$.

For proxy distillation (fine-tuning), we run 20 epochs with inner updates 3 (`FT_INNER_ITER=3`), using AdamW with learning rates $10^{-4}$ for the proxy and $5 \times 10^{-5}$ for the generator. During inference, we employ a parallel stochastic search strategy. Specifically, we generate a batch of $N = 30$ candidate trajectories by injecting Gaussian perturbations ($\sigma = 0.05$) into the initial states. The final counterfactual is derived via a validity-constrained optimization step, where we select the candidate that minimizes the transport cost among all successful trials.

**Implementation of the surrogate** $f_\phi$. In our implementation, $f_\phi$ is realized by two neural proxy classifiers (`C1` and `C2`) that are trained jointly with the generator. Both proxies are $(K+1)$-class networks (`C_OUT_DIM=3` in the binary case): two semantic classes plus an additional noise class for NCE-style density learning. During training, the proxies are updated using cross-entropy on real samples and on uniformly sampled noise points $\tilde{x} \sim \text{Unif}([-c, c]^d)$ labeled as the noise class, with noise ratio $r$ and weight $\lambda_{\text{noise}}$. For generator updates, we use a worst-case aggregation over the two proxies: both the target-class loss and the density score are computed under each proxy and then combined by taking the maximum, which encourages counterfactuals to remain valid and non-noise-like under proxy disagreement. After training, we optionally distill the proxies toward the black-box ensemble by matching the ensemble's mean predicted probability on a small set of generated query points, and continue optimizing the generator using the distilled proxy.

## B.4. Baseline Methods

We summarize the baseline methods and their key assumptions, and describe how we adapt them to our experimental setting for a fair comparison.

- **CEFlow** (Duong et al., 2023) is a robust and efficient counterfactual explanation framework for tabular data using normalizing flows. CEFlow treats mixed-type features (continuous and categorical) via invertible and differentiable flows, enabling fast and stable perturbation addition in latent space. It addresses limitations of VAE-based methods by avoiding sampling randomness, producing more consistent and data-manifold-aligned counterfactuals with improved validity and proximity.

- **Product-MIP** (Leofante et al., 2023) refers to a relational verification approach that constructs a single "product"

neural network combining multiple models under multiplicity, then generates guaranteed robust counterfactuals via mixed-integer programming (MIP). It encodes all equivalent models into one superstructure, enabling exact verification of validity across the model set while analyzing computational complexity for ReLU networks.

- **Argumentative Ensembling** (Jiang et al., 2024a) is a computational argumentation-based approach for robust recourse under model multiplicity. It integrates predictions and counterfactual explanations from multiple equally-performing models simultaneously, using argumentation semantics to aggregate decisions and ensure robustness. The method satisfies desirable properties like counterfactual validity and non-triviality, accommodating user preferences while minimizing accuracy trade-offs in ensembling.

- **BetaRCE** (Stępka et al., 2025) is a post-hoc method providing probabilistic guarantees on counterfactual robustness to model changes. BetaRCE uses a Bayesian-inspired Beta distribution to estimate and shift base counterfactuals toward higher stability regions in feature space, allowing user-defined probability bounds $(\delta, \alpha)$. It operates model-agnostically on top of any base method, preserving plausibility and proximity while outperforming prior robust approaches under natural model updates.

### B.5. Baselines Setting

Here, we show the hyper-parameter settings of the baseline methods used in this paper.

- **Product-MIP** uses an MILP formulation to generate counterfactual explanations robust to model multiplicity. The neural networks consist of two hidden layers with 128 units each and a two-dimensional softmax output layer. The models are optimized using the AdamW optimizer with a learning rate of $10^{-3}$, a batch size of 128, and trained 50 epochs. All numerical features are normalized to $[0, 1]$ and categorical features are label-encoded. Counterfactuals are obtained by minimizing the $\ell_1$ distance subject to MILP constraints, solved using Gurobi.

  **CEFlow** uses a conditional normalizing flow $f_\theta$ to model tabular data and generate counterfactuals via an invertible latent-space perturbation. The flow model is optimized with AdamW using a learning rate of $10^{-3}$, a batch size of 32, and the negative log-likelihood loss for 500 epochs. Following Duong et al. (2023), the hyperparameter $\alpha$ is selected by searching over a range of values; we search over $\alpha \in \{0.5, 0.6, 0.7, 0.8\}$.

- **BetaRCE** uses GrowingSpheres as the generator of base CEs and the fine-tuned black-box ensemble as the 'admissible model space' for its statistical verification algorithm. We run the RCE algorithm with parameter $\beta = [0.5, 0.7, 0.9]$ and distance threshold $\delta = [0.7, 0.8, 0.9]$.

- **Argumentative Ensembling** selects counterfactuals robust to model multiplicity via a bipolar argumentation framework. The well-tuned black-box ensemble $\mathcal{M}$ is used for calling. KDTree nearest-neighbor search (Brughmans et al., 2024) is applied with $k = [3, 5, 7]$, and the BAF semantics are either $s$-preferred or $d$-preferred. Numerical features are scaled to $[0, 1]$ and categorical features are one-hot encoded.

### B.6. Pseudo-Code

## C. Additional Results

### C.1. Endpoint penalty vs. Path penalty

Recalling Sec. 4.1, our framework learns a noise-aware $(K+1)$-way proxy $f_\phi$ that simultaneously supports (i) NCE-based support scoring through an explicit noise class and (ii) differentiable validity signals via local distillation. Given the conditional density score $S(\cdot|y^*)$, we incorporate a rectified density barrier $\text{ReLU}(log\tau - S(\cdot|y^*))$ into the generator objective (Eq. (7)). A natural design question is *whether this penalty should be applied only at the terminal counterfactual (endpoint) or aggregated along the full ODE trajectory (path)?*

Let $z : [0, 1] \to \mathcal{X}$ denote the continuous counterfactual trajectory produced by the ODE generator, with $z(0) = \boldsymbol{x}$ and $z(1) = \boldsymbol{x}'$. We compare two variants that differ only in how the one-sided low-support penalty is evaluated:

---

**Algorithm 1** Training of DensityFlow

---

**Input:** training set $\mathcal{D}$; black-box ensemble $\mathcal{M}$; generator $v_\theta$; surrogate $f_\phi$; training epochs $T$; local distillation epochs $T_{\mathrm{dis}}$.
**Output:** trained generator $v_\theta$ and distilled surrogate $f_{\phi'}$.

1. Initialize the generator parameters $\theta$ and surrogate parameters $\phi$.

2. **Phase I: Surrogate–generator joint training.**

   (a) For $t = 1, \ldots, T$:
       i. Sample a mini-batch $(\boldsymbol{x}, y)$ from $\mathcal{D}$ and choose target labels $y^* \neq y$.
       ii. Update the surrogate $f_\phi$ using the NCE-based objective in Eq. (3).
       iii. Starting from each query $\boldsymbol{x}$, solve the augmented ODE in Eq. (5) to obtain the terminal state $\boldsymbol{x}'$ and the transport cost.
       iv. Update the generator $v_\theta$ using the composite objective in Eq. (7).

3. **Phase II: Query-efficient local distillation.**

   (a) Initialize $f_{\phi'} \leftarrow f_\phi$.
   (b) For $t = 1, \ldots, T_{\mathrm{dis}}$:
       i. Generate terminal states $\boldsymbol{x}' \sim \boldsymbol{z}_\theta(T)$.
       ii. Query the ensemble on these states to obtain $\bar{y} = \mathcal{M}(\boldsymbol{x}')$.
       iii. Form the local support set $\mathcal{D}_\theta = \{(\boldsymbol{x}', \bar{y})\}$.
       iv. Update the local surrogate $f_{\phi'}$ using the distillation loss in Eq. (8).
       v. Refine the generator again using Eq. (7), with $f_{\phi'}$ replacing $f_\phi$.

4. Return $v_\theta$ and $f_{\phi'}$.

---

$$\mathcal{L}_{\mathrm{den}}^{\mathrm{end}}(z) = \mathrm{ReLU}(\gamma - S(z(1)|y^*)) \tag{21}$$

$$\mathcal{L}_{\mathrm{den}}^{\mathrm{path}}(z) = \int_0^1 \mathrm{ReLU}(\gamma - S(z(t)|y^*)) \; dt \tag{22}$$

where $\gamma = \log \tau$ is the margin parameter. $\mathcal{L}_{\mathrm{den}}^{\mathrm{path}}$ is approximated by uniform time discretization with $T$ steps in implementation. All settings were fixed across the two variants, changing only the penalty mode. Following the main paper, we evaluate on the test points that are correctly classified by all black-box models in the ensemble $\mathcal{M}$, generating a counterfactual for each point.

*Table 7.* Endpoint vs. path penalty across datasets.

| Dataset | Cost | | Validity | |
| --- | --- | --- | --- | --- |
| | Endpoint | Path | Endpoint | Path |
| Adult | 1.597±.194 | 1.636±.173 | 0.901±.052 | 0.873±.046 |
| Compas | 1.176±.017 | 1.781±.026 | 0.729±.067 | 0.715±.059 |
| HELOC | 1.812±.599 | 1.824±.524 | 0.757±.035 | 0.744±.028 |
| Blood | 1.527±.401 | 1.519±.318 | 0.662±.046 | 0.718±.084 |

Table 7 reports results on four datasets. Endpoint and path variants are nearly indistinguishable across all metrics. We therefore adopt the endpoint form in the main experiments for computational efficiency. This behavior is consistent with the structure of our objective: the generator is trained under a minimum-action formulation with an explicit kinetic-energy term (Eq. (5)), which favors smooth, short trajectories. At the same time, the noise-aware $(K{+}1)$-way surrogate provides a strong support signal through the conditional score $S(\cdot|y^*)$.

## C.2. Do Density Signals Help? Plug-in Density Proxies vs. NCE

To isolate the role of density information, we compare several plug-in density proxies. For baselines, we define a scalar score $S_{\mathrm{den}}(\boldsymbol{x})$ (where higher values indicate higher density) and use it as a regularizer. Below we define the proxies used in Table 8. Unless stated otherwise, all proxies are computed in the same feature space $\phi(\cdot)$.

---

**Algorithm 2** Inference of DensityFlow for a query instance

---

**Input:** query instance $\boldsymbol{x}$; target label $y^*$; trained generator $v_\theta$; distilled surrogate $f_{\phi'}$; number of trials $N$.
**Output:** counterfactual $\boldsymbol{x}^*$.

1. Construct $N$ perturbed initial states around $\boldsymbol{x}$.

2. For each trial $n = 1, \ldots, N$:

   (a) Solve the augmented ODE in Eq. (5) from the $n$-th initial state.
   (b) Obtain the terminal point $\boldsymbol{x}'_n$ and its transport cost.
   (c) Evaluate whether $\boldsymbol{x}'_n$ satisfies the target validity condition using the distilled surrogate $f_{\phi'}$.

3. Define the valid index set $\mathcal{V} = \{ n \in \{1, \ldots, N\} \mid \arg\max f_{\phi'}(\boldsymbol{x}'_n) = y^* \}$.

4. Return

$$\boldsymbol{x}^* = \begin{cases} \boldsymbol{x}'_{n^*}, & n^* = \arg\min_{n \in \mathcal{V}} cost(\boldsymbol{x}, \boldsymbol{x}'_n), & \text{if } \mathcal{V} \neq \emptyset, \\ \boldsymbol{x}'_{n^*}, & n^* = \arg\min_{1 \le n \le N} cost(\boldsymbol{x}, \boldsymbol{x}'_n), & \text{otherwise.} \end{cases}$$

---

**(i) Discriminator-based proxy (`disc2`).** We train a binary discriminator $d_\psi(\phi(\boldsymbol{x})) \in (0, 1)$ to distinguish in-distribution samples $\boldsymbol{x} \sim \mathcal{D}_{\text{train}}$ from synthetic noise samples $\tilde{\boldsymbol{x}} \sim \mathcal{N}$ (constructed by sampling within the normalized feature range). The density score is defined as

$$S_{\text{den}}^{disc2}(\boldsymbol{x}) = \log \frac{d_\psi(\phi(\boldsymbol{x}))}{1 - d_\psi(\phi(\boldsymbol{x}))} \tag{23}$$

so higher values indicate higher in-distribution likelihood. In practice, this corresponds to the logit of the discriminator output. We implement this discriminator as a lightweight MLP and train it jointly (or as a separate pretraining stage) depending on the experiment.

**(ii) kNN proxy (`knn_reg`).** We compute a local-neighborhood statistic based on $k = 20$ nearest neighbors in $\mathcal{D}_{\text{train}}$:

$$r_k(\boldsymbol{x}) = \frac{1}{k} \sum_{\boldsymbol{x}_j \in \text{kNN}(\phi(\boldsymbol{x}), \phi(\mathcal{D}_{\text{train}}))} \|\phi(\boldsymbol{x}) - \phi(\boldsymbol{x}_j)\|_2 \tag{24}$$

We then map it to a density-like score via a monotone transform, e.g.,

$$S_{\text{den}}^{knn\_reg}(\boldsymbol{x}) = -r_k(\boldsymbol{x}) \tag{25}$$

so larger scores correspond to denser regions (smaller neighbor radius). Our implementation uses `sklearn.neighbors` to query kNN distances.

**(iii) Local Outlier Factor proxy (`lof_reg`).** We use the LOF score $\text{LOF}(\boldsymbol{x})$ computed against $\mathcal{D}_{\text{train}}$:

$$S_{\text{den}}^{\text{lof\_reg}}(\boldsymbol{x}) = -\text{LOF}(\phi(\boldsymbol{x})) \tag{26}$$

where larger values indicate less outlierness (higher local density). We compute LOF using the standard implementation in `sklearn.neighbors.LocalOutlierFactor`.

Finally, we incorporate these signals into the main optimization objective. For the plug-in baselines (where a higher $S_{\text{den}}$ implies higher density), the penalty takes the form:

$$\mathcal{L}_{\text{den}}(\boldsymbol{x}) = \max\big(0, log\tau - S_{\text{den}}(\boldsymbol{x})\big) \tag{27}$$

For our method, we use the proposed score $S(\boldsymbol{x}|y^*)$ with the penalty $\text{ReLU}\big(\log\tau - S(\boldsymbol{x}|y^*)\big)$.

Table 8 shows that density signal generally improves validity compared to using no density regularization (`None`), indicating that discouraging low-density regions is beneficial for CE generation. However, off-the-shelf plug-in proxies (e.g., `disc2`, `KNN`, and `LOF`) exhibit noticeable dataset dependence. In contrast, the proposed conditional score $S(\cdot|y^*)$ (labeled as NCE) achieves the best validity on three out of four datasets and is overall more consistent. This suggests that a learned density-ratio signal tailored to the target class provides more reliable optimization guidance than generic neighborhood/outlier heuristics. Moreover, our method is naturally differentiable and incurs negligible overhead as it reuses the trained classifier, whereas KNN and LOF require nearest-neighbor searches at each ODE step.

*Table 8.* Performance comparison of validity across datasets with different density estimators.

| Dataset | None | Disc2 | KNN | LOF | NCE (ours) |
|---------|------|-------|------|------|------------|
| Adult | 0.815 | 0.823 | 0.878 | 0.886 | 0.901 |
| Blood | 0.495 | 0.424 | 0.560 | 0.696 | 0.662 |
| Compas | 0.642 | 0.689 | 0.645 | 0.720 | 0.729 |
| HELOC | 0.718 | 0.728 | 0.681 | 0.742 | 0.757 |

## C.3. Consistency of NCE with other standard density estimation methods

We validate whether our NCE-based density proxy (implemented via the density score) is consistent with widely used density/support estimators on real tabular datasets. Concretely, we train a lightweight NCE discriminator to separate in-distribution samples from synthetic "noise" samples, and use the resulting score to quantify how noise-like a point is.

To assess agreement with classical density estimators, we compute Spearman's rank correlation $\rho$ between the NCE density proxy and each baseline score on the test set. We focus on Spearman correlation because different estimators have incomparable scales, and rank consistency is the most meaningful notion of agreement across methods.

We compare against six standard density/support estimators spanning both local and global notions of support: (i) one-class SVM (OC-SVM), (ii) PCA+KDE (kernel density estimation after PCA), (iii) Mahalanobis distance under a global Gaussian model, (iv) Isolation Forest, (v) Elliptic Envelope (robust covariance), and (vi) a kNN radius-based log-density proxy. All baseline scores are oriented so that higher values indicate more in-distribution (denser) points.

As shown in Table 9, the NCE density proxy displays meaningful rank-level agreement with standard estimators. While the best-matching baseline varies by dataset, the correlations remain significant throughout. The strongest agreement typically occurs with methods that capture neighborhood structure, such as kNN and PCA+KDE, or coarse global geometry like Mahalanobis distance. Overall, these results suggest that the NCE proxy recovers a broadly compatible notion of in-distribution support, providing empirical justification for using it as a practical surrogate to identify low-support (tail) regions that are most relevant to robustness and boundary uncertainty.

*Table 9.* NCE density proxy and classical density estimators on real datasets (Spearman's $\rho$; higher is better).

| Dataset | OC-SVM | PCA+KDE | Mahalanobis | IsoForest | EllipticEnv | kNN |
|---------|--------|---------|-------------|-----------|-------------|------|
| Adult | 0.2558 | 0.4015 | 0.5067 | 0.4317 | 0.3425 | 0.5276 |
| Compas | 0.8226 | 0.7348 | 0.6757 | 0.6301 | 0.6026 | 0.5994 |
| HELOC | 0.3957 | 0.2884 | 0.5673 | 0.5481 | 0.4891 | 0.3949 |
| Blood | 0.1296 | 0.6935 | 0.1933 | 0.4540 | 0.5088 | 0.6513 |

## C.4. Trade-off between Cost and Validity

We examine how the objective weights on validity and cost shape the CE solutions. We sweep $\lambda_{\text{valid}} \in \{1, 1.25, 1.5, 1.75, 2\}$ and $\lambda_{\text{cost}} \in \{0.2, 0.3, 0.4, 0.5, 0.6\}$ while keeping other training and inference settings fixed.

Across datasets, the sweep shows a consistent Pareto trade-off between validity and cost. Increasing $\lambda_{\text{cost}}$ systematically reduces $\ell_2$ cost, typically at the expense of validity: stronger cost regularization discourages large moves needed to cross the decision boundary, reducing the flip rate. Conversely, increasing $\lambda_{\text{valid}}$ shifts solutions toward higher validity, usually with higher $\ell_2$ cost, reflecting a stronger incentive to prioritize boundary crossing over proximity. For a fixed $(\lambda_{\text{valid}}, \lambda_{\text{cost}})$, the spread induced by different `lambda_den` values is generally smaller than the shifts caused by changing $\lambda_{\text{valid}}$ or $\lambda_{\text{cost}}$, supporting a clean separation of Density-related effects in a standalone experiment. Overall, the resulting scatter forms the expected trade-off surface: moving toward higher validity generally requires paying a higher cost, and vice versa. A practical operating point can be chosen near the "knee" of this curve, where most of the validity gains are retained without incurring the steepest cost increase (often around mid-range $\lambda_{\text{valid}}$ with $\lambda_{\text{cost}} = 0.4$ in our sweeps).

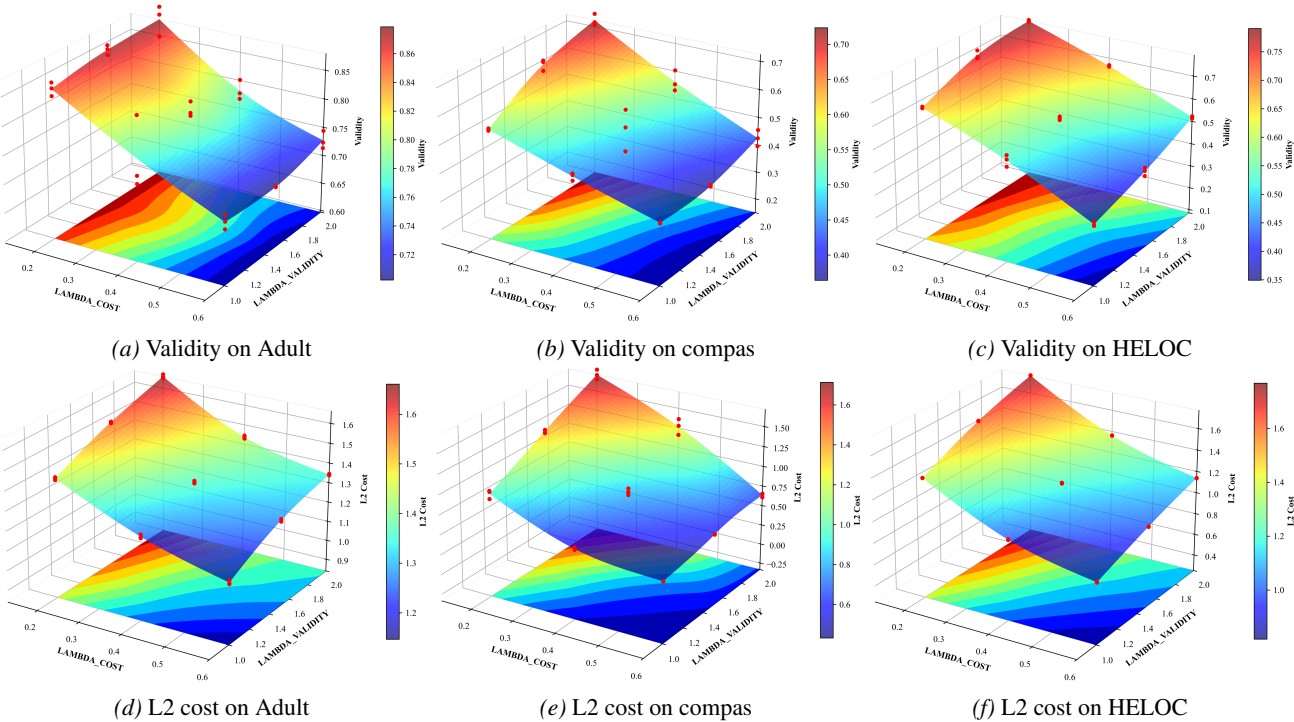

*(a) Validity on Adult*  *(b) Validity on compas*  *(c) Validity on HELOC*

*(d) L2 cost on Adult*  *(e) L2 cost on compas*  *(f) L2 cost on HELOC*

*Figure 9.* Cost–validity trade-off surfaces under varying objective weights. We visualize the validity (top row) and $\ell_2$ cost (bottom row) of generated counterfactual explanations as functions of the objective weights $\lambda_{\text{valid}}$ and $\lambda_{\text{cost}}$ on the Adult, Compas, and HELOC datasets.

### C.5. Discussion of Noise (Ratio and Range)

**Detailed Analysis.** In Section 3.1, we defined the trust region threshold relative to the reference noise. In our NCE implementation, we do not perform complex quantile estimation. Instead, the boundary is controlled directly by two hyperparameters: the noise ratio $\tau$ (which sets the strictness) and the clipping range $C$ which sets the noise density. Here, $\tau$ corresponds strictly to the sampling ratio $N_{\text{noise}}/N_{\text{data}}$. We derive this relationship and analyze the impact of $C$.

The NCE discriminator $f_\phi$ learns to distinguish between the data distribution $p_{\text{data}}(\boldsymbol{x})$ and the noise distribution $p_{\text{noise}}(\boldsymbol{x})$. The implicit decision boundary (where $f_\phi$ outputs 0.5) converges to the level set:

$$p_{\text{data}}(\boldsymbol{x}) = \tau \cdot p_{\text{noise}}(\boldsymbol{x}) \tag{28}$$

In our framework, noise is sampled uniformly from a hypercube of side length $2C$ in $d$ dimensions. Thus, the noise density is constant:

$$p_{\text{noise}}(\boldsymbol{x}) = \frac{1}{\text{Volume}} = \frac{1}{(2C)^d} \tag{29}$$

Consequently, the implicitly learned density threshold is $\tau \cdot (2C)^{-d}$. The corresponding rejection quantile $\alpha$ (the proportion of data excluded from the trust region) is strictly a function of $C$:

$$\alpha(C) = \int_{\{\boldsymbol{x}: p_{\text{data}}(\boldsymbol{x}) < \tau(C)\}} p_{\text{data}}(\boldsymbol{x}) d\boldsymbol{x} \tag{30}$$

This derivation proves that controlling the geometric parameter $C$ is mathematically equivalent to controlling the statistical threshold $\alpha$. As the noise volume $(2C)^d$ increases, the threshold decreases, leading to a smaller $\alpha$. A small $C$ implies a large $\alpha$, causing the model to reject difficult samples (Survivorship Bias).

**Experiments.** Our NCE-based density score $S(\boldsymbol{x} \mid y^*)$ is learned by augmenting the $K$ data classes with an additional uniform-noise class. This construction introduces two practical hyperparameters: (i) the noise ratio ($\tau$), i.e., the fraction of noise samples used to train the $(K{+}1)$-way discriminator; and (ii) the noise sampling range ($C$), implemented by drawing uniform noise within a clipped box in the feature space.

*Table 10.* Sensitivity analysis of NCE hyperparameters. Cost($\downarrow$) and Validity ($\uparrow$) are reported with mean$\pm$std over 5 different seeds.

| Parameter | Value | Adult | | Compas | |
|---|---|---|---|---|---|
| | | Cost | Validity | Cost | Validity |
| $\tau$ (Ratio) | 0.1 | $1.529 \pm .178$ | $0.859 \pm .116$ | $1.172 \pm .022$ | $0.682 \pm .102$ |
| | 0.2 | $1.597 \pm .194$ | $0.901 \pm .052$ | $1.235 \pm .020$ | $0.729 \pm .067$ |
| | 0.3 | $1.580 \pm .165$ | $0.887 \pm .049$ | $1.294 \pm .018$ | $0.710 \pm .046$ |
| | 0.5 | $1.533 \pm .144$ | $0.873 \pm .044$ | $1.318 \pm .027$ | $0.757 \pm .082$ |
| | 1.0 | $1.507 \pm .175$ | $0.895 \pm .042$ | $1.363 \pm .059$ | $0.744 \pm .044$ |
| $C$ (Range) | 0.2 | $3.742 \pm .556$ | $0.662 \pm .187$ | $0.694 \pm .024$ | $0.567 \pm .147$ |
| | 0.5 | $2.934 \pm .603$ | $0.727 \pm .103$ | $1.054 \pm .023$ | $0.653 \pm .091$ |
| | 1.0 | $1.857 \pm .210$ | $0.892 \pm .032$ | $1.093 \pm .015$ | $0.684 \pm .036$ |
| | 2.0 | $1.355 \pm .108$ | $0.883 \pm .038$ | $1.180 \pm .016$ | $0.726 \pm .055$ |
| | $\beta \cdot \max_{x \in \mathcal{D}_{\text{train}}} \|x\|_\infty$ | $1.597 \pm .194$ | $0.901 \pm .052$ | $1.176 \pm .017$ | $0.729 \pm .067$ |

To assess sensitivity, we vary one factor at a time while keeping all other settings fixed. Table 10 shows that DensityFlow is not very sensitive to `noise_ratio`, except that overly small ratios (e.g., 0.1) can hurt performance due to insufficient noise samples for learning a stable NCE signal. Regarding the noise range $C$, narrow ranges substantially degrade validity, while broader ranges improve results. This ensures that the noise distribution covers the entire support of the data with a small margin (buffer zone).

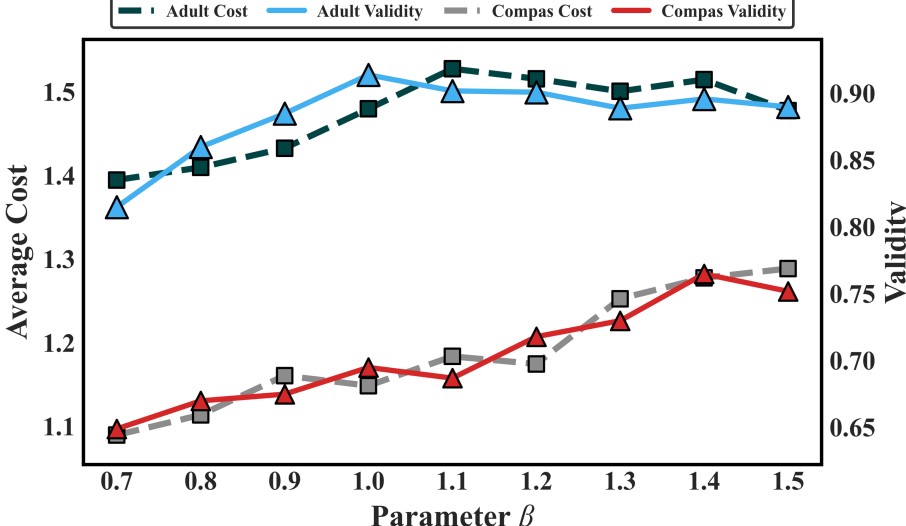

*Figure 10.* Sensitivity of $\beta$. We report both Cost (solid) and Validity (dashed). As $\beta$ increases, both metrics stabilize at a robust plateau.

**Impact of the noise bound $C$.** We analyze the sensitivity of the noise range $C$, which is controlled by the factor $\beta$ via $C = \beta \cdot \max_x \|x\|_\infty$. $C$ alters the noise density exponentially, while the noise ratio $\tau$ scales the density threshold linearly in Appendix C.5. As shown in Fig. 10, initially, the low metrics at strict settings ($\beta < 0.9$) are artifacts of excluding difficult samples. As $\beta \in [1.0, 1.4]$, the cost rises to reflect the "cost of robustness", forcing the generator to traverse the trust regions. Validity also stabilizes at a high plateau, indicating that DensityFlow is robust to $\beta$ variations and insensitive to precise tuning.

## C.6. Ensemble Agreement Along the Trajectory

Here, we include an intuition visualization. For test instances, we generate counterfactual trajectories with our DensityFlow and sample points at $T \in \{0, 0.25, 0.5, 0.75, 1\}$. At each $T$, we query the black-box ensemble and report the fraction of samples whose ensemble majority vote predicts class 1 (blue) versus class 0 (red), shown as a stacked bar plot over $T$.

Fig. 11 exhibits a consistent "source-side consensus $\rightarrow$ mid-trajectory transition $\rightarrow$ target-side consensus" pattern: predictions are largely stable near $T = 0$, become more mixed around intermediate $T$ (reflecting the boundary-crossing transition

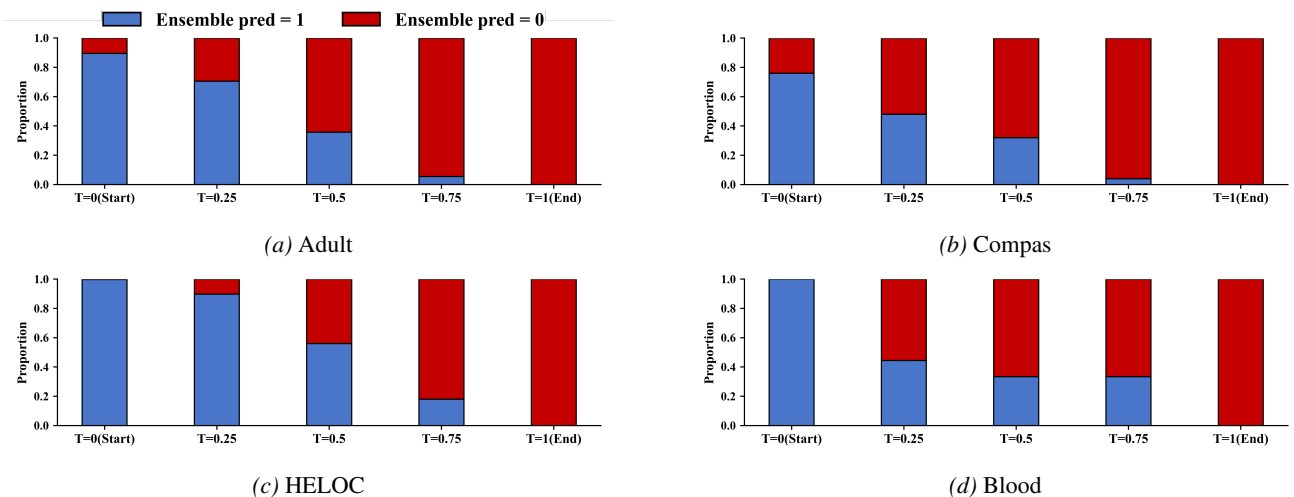

*Figure 11.* Ensemble majority-vote label proportions along the counterfactual trajectory generated by our flow model. For each dataset, we sample points at $T \in \{0, 0.25, 0.5, 0.75, 1\}$ and report the fraction predicted as class 1 (blue) vs. class 0 (red), illustrating a consensus–transition–consensus pattern from start to end.

band), and re-stabilize near $T = 1$. This provides a compact visual sanity check for our narrative that counterfactual generation traverses an unstable region near the decision boundary before settling in a more stable region.

### C.7. Density Score vs. Ensemble Disagreement

We empirically assess Assumption A.2 by testing whether our learned density score identifies regions with elevated ensemble (epistemic) disagreement. We use the class-conditional density score $S(\boldsymbol{x} \mid y^*)$, where larger values indicate *higher* local data density. Ensemble disagreement is quantified by $\mathrm{Var}(p) = \mathrm{Var}_{m=1}^{M}\big(p_m(y^* \mid \boldsymbol{x})\big)$, computed from per-model predicted probabilities queried from the ensemble $\mathcal{M}$.

On test points, we report (i) the Spearman correlation between *negative* density and disagreement, i.e., $\rho_s(-S(\boldsymbol{x} \mid y^*), \mathrm{Var}(p))$, and (ii) a tail concentration metric:

$$\mathrm{Tail} = \frac{\mathbb{E}[\mathrm{Var}(p) \mid S(\boldsymbol{x} \mid y^*) \in \text{bottom 20\% (lowest density)}]}{\mathbb{E}[\mathrm{Var}(p) \mid S(\boldsymbol{x} \mid y^*) \in \text{top 20\% (highest density)}]}$$

To probe a controlled density shift, we construct an interpolation path $x(\alpha) = (1 - \alpha)x_{\mathrm{real}} + \alpha x_{\mathrm{noise}}$ and report the end-to-end increase:

$$\Delta\mathrm{Var}_{0\to1} = \mathbb{E}[\mathrm{Var}(p) \mid \alpha = 1] - \mathbb{E}[\mathrm{Var}(p) \mid \alpha = 0]$$

This interpolation serves as a controlled density-shift probe and is not intended to preserve actionability. $[\cdot, \cdot]$ denote bootstrapped 95% confidence intervals. Stronger alignment between *lower* density and higher disagreement corresponds to: (i) positive $\rho_s(-S(\boldsymbol{x} \mid y^*), \mathrm{Var}(p))$, (ii) Tail $> 1$, and (iii) positive $\Delta\mathrm{Var}_{0\to1}$.

*Table 11.* Key statistics for whether lower density (measured by $-S(\boldsymbol{x} \mid y^*)$) aligns with higher ensemble disagreement.

| Dataset | $\rho_s(-S(\boldsymbol{x} \mid y^*), \mathrm{Var}(p))$ | Tail | $\Delta\mathrm{Var}_{\alpha:0\to1}$ |
|---|---|---|---|
| Adult | .207 [.187,.259] | 2.024 [1.571,2.649] | .033 [.031,.037] |
| Compas | .238 [.195,.286] | 1.626 [1.363,1.956] | .017 [.015,.018] |
| HELOC | .216 [.171,.258] | 1.793 [1.592,2.039] | .026 [.024,.027] |
| Blood | .187 [.125,.337] | 1.556 [1.121,2.056] | .012 [.011,.014] |

Table 11 shows consistent alignment between lower density and higher model disagreement: $\rho_s(-S(\boldsymbol{x} \mid y^*), \mathrm{Var}(p))$ is positive with 95% CIs excluding 0, Tail $> 1$ indicates disagreement concentrates in the low-density tail, and $\Delta\mathrm{Var}_{0\to1} > 0$ confirms disagreement increases along the real-to-noise (lower-density) interpolation.

## C.8. Running Time Comparison

Here we report the end-to-end runtime of the full pipeline. Since the compared baselines are quite different in nature(e.g., search-based, generation-based, optimization-based), a unified theoretical complexity comparison would be less informative. The results in Table 12 show that DensityFlow is slightly more expensive than some simpler baselines, which is expected given the additional modules. However, this cost is accompanied by substantially better robustness under MM.

*Table 12.* Running time comparison (in seconds)

| Dataset/Method | Product_mip | CeFlow | BetaRCE | Argument | DensityFlow |
|---|---|---|---|---|---|
| Spirals | 704.7 | 394.6 | 747.9 | 205.7 | 655.6 |
| HELOC | 335.1 | 881.5 | 427.3 | 647.7 | 734.0 |

## C.9. Evaluation on External Plausibility Metrics

Following prior works (Leofante et al., 2023), we add experiments on external plausibility metrics, LOF (Breunig et al., 2000) and Isolation Forest (Liu et al., 2008). The results in Table 13 show that DensityFlow achieves better plausibility under both external detectors. At the same time, we would like to clarify the role of these metrics in our setting. Under MM, our primary question is whether a CE remains stably valid across a set of plausible models. Therefore, LOF or Isolation Forest can serve as useful external plausibility indicators, but they should not be treated as the gold standard for evaluation.

*Table 13.* Performance comparison on two plausibility metrics LOF($\downarrow$) | ISOFOREST($\downarrow$).

| Dataset/Method | Product_mip | CeFlow | BetaRCE | Argument | DensityFlow |
|---|---|---|---|---|---|
| Spirals | 1.29 \| 0.53 | 1.25 \| 0.59 | 1.13 \| 0.49 | 1.16 \| 0.50 | **1.07** \| **0.48** |
| HELOC | 1.54 \| 0.44 | 2.59 \| 0.48 | 3.17 \| 0.57 | 1.07 \| **0.42** | **1.04** \| **0.42** |

## C.10. Scalability to High-Dimensional Data

Both density estimation and continuous flow generation face inherent challenges when scaled to high-dimensional spaces due to the 'curse of dimensionality'. In this part, we explore a primary direction to scale DensityFlow to high-dimensional data and discuss their respective advantages and limitations.

A straightforward approach to mitigate dimensionality is feature selection (FS). FS aims to identify the most relevant features for the prediction target to build more interpretable and robust models (Tan et al., 2024). We designed an FS-variant (DensityFlow-FS) that employs LassoNet (Lemhadri et al., 2021) to identify the Top-K features most relevant to the target label. Only these selected features are fed into the continuous flow generation, while the remaining features are kept frozen. Table 14 shows the performance of DensityFlow-FS on the Musk dataset (6598, 169) across varying $K$. The results show that extracting the Top-5 or Top-10 features achieves strong validity and low transport cost. However, as $K$ increases to 20, validity slightly drops while the cost increases, suggesting that filtering out irrelevant dimensions effectively reduces noise and stabilizes the ODE trajectory.

*Table 14.* Performance of *DensityFlow-FS* on the Musk dataset with different numbers of selected features.

| Method | Cost ($\downarrow$) | Validity ($\uparrow$) |
|---|---|---|
| DensityFlow-FS (Top-5) | 8.57 | 0.86 |
| DensityFlow-FS (Top-10) | 8.73 | 0.85 |
| DensityFlow-FS (Top-20) | 10.26 | 0.80 |

**Failure Mode.** While feature selection is pragmatic, it fundamentally assumes that the discarded 'non-predictive' features are independent of the modified predictive features. As noted in the main text's limitations, this assumption breaks down if a causal or physical coupling exists. For instance, in a water quality assessment dataset, a feature like 'conductivity' might lack predictive power for potability and thus be discarded by LassoNet. If our framework drastically alters the 'pH' level to achieve a counterfactual flip, the physical environment demands a corresponding shift in conductivity. Leaving the

coupled non-predictive feature unchanged results in a physically impossible, out-of-distribution (OOD) sample (e.g., 'high pH but near-zero conductivity'). Resolving this requires integrating causal graphs or structural causal models into the flow constraints, which remains a promising direction for future work.

## C.11. Discussion on Counterfactual Diversity

In this section, we discuss the diversity of DensityFlow. Methodologically, two designs in DensityFlow prevent it, to a certain extent, from collapsing into a single class prototype or a global density mode: **(i)** Inference-time parallel stochastic search. We generate a batch of candidate trajectories by injecting Gaussian perturbations into the initial query. This stochastic initialization forces the velocity field generator to explore distinct pathways across the data manifold. **(ii)** Balanced composite objective. Our optimization objective Eq. (7) balances the density potential and transport cost. Consequently, the model generally does not blindly chase the absolute highest-density regions of the target class.

Here, we conduct a qualitative analysis on the Adult dataset. In the training set, high-income ($\geq$ 50K) exhibit a severe gender imbalance, where females constitute a small minority subgroup (only $15\%$) compared to males ($85\%$).

We sample test instances initially labeled as 'income $<$ 50K' with an equal male-to-female ratio ($50/50$) as queries, and generate CEs for target '$\geq$ 50K'. As summarized in Table 15, while the generated CEs show a mild, natural shift toward the dominant modality, a substantial proportion ($43.02\%$) of the female queries successfully find valid recourse strictly within the female minority subgroup, instead of being forced or corrupted into the dominant male mode.

*Table 15.* Comparison of sex feature distribution between original queries and generated CEs on the Adult dataset.

| Phase | Male ($\%$) | Female ($\%$) |
|---|---|---|
| Queries | 50.09% | 49.91% |
| Robust CEs | 56.98% | 43.02% |

Table 16 presents an example from this minority subgroup. The query is a 19-year-old female with a high school education, working 12 hours per week. The counterfactual explanation updates her profile to a 30-year-old female working full-time (40 hours per week) in a professional role within the federal government. This modification aligns with general intuition, demonstrating that the generated direction is typically reasonable.

*Table 16.* Qualitative counterfactual recourse example for a female instance in the minority subgroup.

| Feature | Original Sample | Counterfactual |
|---|---|---|
| Age | 19.0 | 30.0 |
| Workclass | Private | Federal-gov |
| Education | HS-grad | Bachelors (or Specialty) |
| Marital Status | Never-married | Married-civ-spouse |
| Occupation | Other-service | Prof-specialty |
| Relationship | Own-child | Wife |
| Race / Sex | White Female | White Female |
| Hours per week | 12.0 | 40.0 |
| **Income** | **$<$ 50K** | **$\geq$ 50K** |

## D. Visualization on Synthetic Datasets

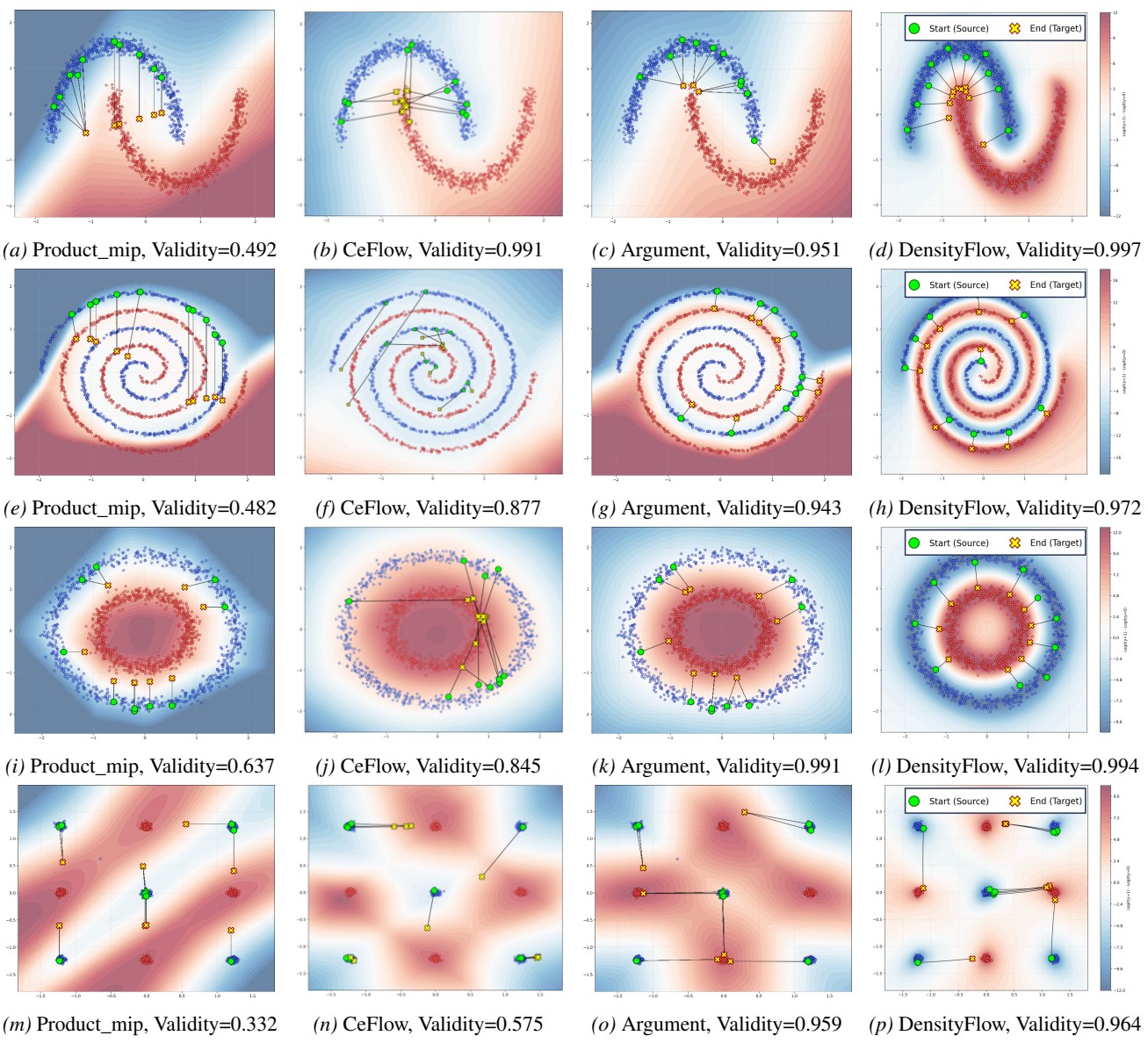

*(a)* Product_mip, Validity=0.492  *(b)* CeFlow, Validity=0.991  *(c)* Argument, Validity=0.951  *(d)* DensityFlow, Validity=0.997

*(e)* Product_mip, Validity=0.482  *(f)* CeFlow, Validity=0.877  *(g)* Argument, Validity=0.943  *(h)* DensityFlow, Validity=0.972

*(i)* Product_mip, Validity=0.637  *(j)* CeFlow, Validity=0.845  *(k)* Argument, Validity=0.991  *(l)* DensityFlow, Validity=0.994

*(m)* Product_mip, Validity=0.332  *(n)* CeFlow, Validity=0.575  *(o)* Argument, Validity=0.959  *(p)* DensityFlow, Validity=0.964

*Figure 12.* Visualization results on four synthetic datasets.

