# OpenReview forum: "Density-Guided Robust Counterfactual Explanations on Tabular Data under Model Multiplicity"
_ICML.cc/2026/Conference — ICML 2026 regular_

### Official Review · Reviewer_yeSp · 2026-02-16

**Soundness:** 3
**Presentation:** 3
**Significance:** 3
**Originality:** 2
**Overall Recommendation:** 5
**Confidence:** 3

**Summary:**

The paper introduces "DensityFlow", a new robust counterfactual explainer for ensemble predictors on tabular data under model multiplicity. For this, they use a density-guided neural ODE, which uses continuous flow dynamics to avoid low-density areas. The low-density areas are avoided by using a differentiable noise contrastive estimation of the density score. Moreover, they designed a trajectory-aware distillation improving query speed for models that only provide predictions (in their words, "Black-box models").

**Compliance With Llm Reviewing Policy:**

Affirmed.

**Final Justification:**

The authors did a really strong rebuttal, which made the paper clearer to me and could resolve my primary concern that they sacrifice diversity for the sake of counterfactual quality.

**Key Questions For Authors:**

1) Why is there no comparison or citation of highly cited counterfactual methods for tabular counterfactuals like DiCE [1] etc? I understand that the focus is slightly different, but e.g. CeFlow also directly compares with it.
2) The general framing is quite confusing…why do black-box models not have gradients? Usually, in the most cited explainable machine learning papers like [2], neural networks are considered black boxes despite having gradients…I am personally not that experienced with tabular data and have more experience with counterfactuals for image/language models, but many models for tabular data are also differentiable and one could also use neural networks (like recently hyped TabPFN) for it…when reading it more closely I realize that you are trying to explain diverse Ensemble models under model multiplicity only, but I think that this should be clarified latest in the abstract to clearly give an idea about the scope of the paper. I actually think it should even be already in the title to not give a wrong impression about the scope of the paper as I had it for most of the time when reading it
3) Regarding figure 1: what space is shown there? Ambient or latent space? If it was actually in ambient space as it seems to me, the idea that directly optimizing there seems to me long outdated, latest [3] in 2021 has shown that one should optimize in latent space if one can’t formulate explicit criteria (like DiCE or mixed linear integer programming done in tabular counterfactuals) to avoid adversarial attacks or other OOD counterfactuals and using diffusion for counterfactual explanations has also become the norm since 2022 with DiME [4] - both are also not cited and to compare against the absolute trivial baseline of just naively optimizing in ambient space seems like a strawman argument to me
4) Regarding Figure 2:  Using distillation for better counterfactuals for the sake of quality or also query speed is not really new; in visual counterfactuals, e.g., SCE [5] and TIME [6] do it as well…I do not fully understand your motivation, though. I see the point of problematic low-density areas (or I would, in my words, call it disconnected modes), but how does distillation help with this problem? Aren’t the two problems that a) you have no gradients in your model and b) that the class modes are disconnected, not independent problems? Generally, I think there are too many things in Figure 2, and I can’t fully understand what the point of them is…I do not even fully understand what kind of diagram this is - am I seeing some kind of training or inference flow here? Where are the actual counterfactuals coming out?
5) Regarding the general logic: why is it desirable to keep counterfactuals in high-density areas only? I understand keeping them inside the support on manifold, but if I forbid them from going into low-density areas, am I not biasing my explanation by this? E.g., when trying to find problems in the model and fix them as with CFKD [7], the most interesting counterfactuals are the ones that are in small subgroups of the distribution, because in this case, you can effectively find problems like shortcut features in your classifier. What point does it have to always only show high-density counterfactuals, which usually only show you what you already knew before?
6) Why are the reported metrics different from, e.g., the CeFlow paper, which evaluates on the same datasets, but they report very different metrics
7) Why is there no comparison or citation of “Probabilistically Plausible Counterfactual Explanations with Normalizing Flows (PPCEF)” [8]? Both methods treat the counterfactual generation process as an optimization problem where the loss function balances validity, proximity, and plausibility.

[1] Mothilal, R.K., Sharma, A. and Tan, C., 2020, January. Explaining machine learning classifiers through diverse counterfactual explanations. In Proceedings of the 2020 conference on fairness, accountability, and transparency (pp. 607-617).
[2] Ribeiro, Marco Tulio, Sameer Singh, and Carlos Guestrin. "" Why should i trust you?" Explaining the predictions of any classifier." Proceedings of the 22nd ACM SIGKDD international conference on knowledge discovery and data mining. 2016.
[3] Dombrowski, Ann-Kathrin, Jan E. Gerken, and Pan Kessel. "Diffeomorphic explanations with normalizing flows." ICML workshop on invertible neural networks, normalizing flows, and explicit likelihood models. 2021.
[4] Jeanneret, Guillaume, Loïc Simon, and Frédéric Jurie. "Diffusion models for counterfactual explanations." Proceedings of the Asian conference on computer vision. 2022.
[5] Bender, Sidney, et al. "Towards desiderata-driven design of visual counterfactual explainers." Pattern Recognition (2025): 112811.
[6] Jeanneret, Guillaume, Loïc Simon, and Frédéric Jurie. "Text-to-image models for counterfactual explanations: a black-box approach." Proceedings of the IEEE/CVF Winter Conference on Applications of Computer Vision. 2024.
[7] Bender, S., Anders, C. J., Chormai, P., Marxfeld, H. A., Herrmann, J., & Montavon, G. (2023). Towards fixing clever-hans predictors with counterfactual knowledge distillation. In Proceedings of the IEEE/CVF International Conference on Computer Vision (pp. 2607-2615).
[8] Wielopolski, Patryk, et al. "Probabilistically plausible counterfactual explanations with normalizing flows." arXiv preprint arXiv:2405.17640 (2024).

**Limitations:**

I think a core limitation that was not mentioned is that I am not convinced at all that the core goal they are following, restricting counterfactuals to high-density areas only, is actually in any way useful. From my point of view, the point of explainable machine learning is finding edge cases, spurious correlations, and unexpected behaviour, and this method intentionally biases the explanation so that the used metrics look good, but show no guarantee that the explanation is in any way useful for an actual application.

**Strengths And Weaknesses:**

Strengths:
+ the math and the methodology look sound to me
+ on the selected metrics the method beats related work

Weaknesses:
- i see no proof that the goal to stay in high-density areas with the counterfactual explanations actually is useful for an actual application and not only for optimizing the selected metrics
- a lot of related work seems to be omitted ranging from similiar or foundational work done many years ago in computer vision (like DiME or TIME) to the most highly cited papers in counterfactuals for tabular data such as DiCE
- I personally think the figures are very confusing (figure 1 shows something that is long known and does not say much about the actual contribution of the paper and figure 2 contains too many different elements without clearly showing how they actually work together)
- given that ICML is a general broad machine learning conference terminology like "black box model" should be explained carefully especially when using them differently than the most cited papers in explainable machine learning
- title and abstract should more clearly state that this method is specific for ensemble predictors on tabular data under model multiplicity

---

> ### Author Rebuttal · Authors · 2026-03-30
>
> Thank you for the detailed and candid feedback. We‘ll address your main concern below and will revise the manuscript.
>
> **Q1. 'Black-box' scope and the scope of the paper need clarity**
>
> **A1.** We appreciate the feedback. We have changed the title to "**Density-Guided Robust Counterfactual Explanations on Tabular Data under Model Multiplicity**" and revised the abstract to state the problem setting upfront. We have also clarified our usage of 'black-box model': it means the target system can only be accessed through prediction queries, while internal constituent models, their parameters, and gradients are unavailable. We thank you for this suggestion.
>
> **Q2. Missing citation/comparison to classical/related methods**
>
> **A2.** Thank you for your suggestion. DiCE/PPCEF were not selected because they focus on single-model explanation, which differs from MM and makes direct comparison hard. We will cite and discuss them in the revised paper.
>
> To address your concerns, we design a forced comparison method that involves performing the comparison on a well-tuned MLP to obtain x', and then feeding x' into an ensemble to evaluate its effectiveness. Table 1 shows that they generally have poor effectiveness under the MM evaluation protocol, which might be due to the discrepancy in objectives.
>
> Table 1. Comparison results with DiCE & PPCEF
> |Datasets|Method|Cost↓|Validity↑|
> |---|---|---|---|
> |Spirals|DensityFlow|0.48|0.97|
> ||DiCE|0.49|0.76|
> ||PPCEF|0.71|0.81|
> |HELOC|DensityFlow|1.81|0.76|
> ||DiCE|7.02|0.16|
> ||PPCEF|1.54|0.47|
>
> **Q3.  Fig.1 and 2 aren't clear enough, distillation isn't new**
>
> **A3.** Thanks for pointing this out. Fig.1 is only **a conceptual illustration**: low-density regions are where plausible models tend to disagree more, so a CE that lands there is less likely to remain robust across models. Our RQ2 and Appx. C.7 also show that lower density corresponds to higher ensemble disagreement.
>
> We would like to clarify that our framework is not restricted to feature(ambient) space, the density-guided ODE is space-agnostic and can operate in latent space. We design a latent-variant(*-L) using a β-VAE[1]. Table 2 shows that DensityFlow transfers effectively. That said, ambient-space methods remain common in tabular CE due to the heterogeneous nature of tabular features and their interpretable semantics. We'll cite [3,4] you mentioned and strengthen the discussion of latent-/diffusion-based related work.
>
> Table 2. Ambient space vs. latent space on Adult
> |Method|Cost↓|Validity↑|
> |---|---|---|
> |DensityFlow|1.59|0.90|
> |*-L|1.53|0.89|
>
> [1]Zhang H, et al. Mixed-type tabular data synthesis with score-based diffusion in latent space
>
> **Fig.2:** We agree that Fig.2 is too complex. We'll simplify it by separating (i) the main generation path from (ii) the distillation loop that aligns the surrogate with the queried ensemble. The final output is z(T), which is mapped back to x' by the inverse transform.
>
> We agree that the concept of distillation isn't new, but distillation with the density guide is new. About TIME & SCE you mentioned, they use distillation in different ways: TIME distills class semantics into the generative model; SCE distills the original model into a smoother surrogate for stabler gradients. In contrast, we use distillation for trajectory-aware local alignment, matching the surrogate to the ensemble near the density-guided path.
>
> **Q4. Why prefer higher-density CEs? Would this miss edge but interesting cases?**
>
> **A4.** We believe _edge but interesting CEs_ and _robust CEs_ are two **totally different domains** that naturally align with different regions of the data distribution. While edge cases often reside in low-density to provide atypical insights, robust CEs need to prioritize high-density to ensure validity across multiple models. One could discover edge cases by adding a diversity-promoting prior to the density score to explore sparse regions. This is a promising direction for future work, thank you for this insightful suggestion. We'll cite CFKD and make this boundary explicit in the limitations section.
>
> Regarding concerns about diversity of high-density CEs, our objective also includes cost & path constraints. Therefore, our method doesn't collapse the CEs to class prototypes or to the globally densest region. Instead, it mainly suppresses isolated target points with weak local support, because such points are often unstable across models. When both validity and cost allow, subgroup targets with good local support can still be reached.
>
> **Q5. Why the reported metrics differ from CeFlow**
>
> **A5.** In our paper, baselines are evaluated under the MM protocol, we report ensemble validity and a unified cost metric. We also evaluated our method in a single model protocol, DensityFlow achieves near-perfect validity. Metrics such as mean log-density in CeFlow are designed for a single model. Based on Reviewer rEzf's comments, we added external plausibility metrics; please refer to **A4**.

---

> > ### Author Rebuttal · Reviewer_yeSp · 2026-03-31
> >
> > I appreciate your updated experiments and think that you improved clarity already a lot. I am willing to increase my score to accept if you show me some convincing qualitative counterfactuals created by your method for  e.g. the adult or compass dataset which are actually falling into a minority group (like high income females on adult) or point out some problematic behaviour in another way.

---

> > > ### Author Response · Authors · 2026-04-02
> > >
> > > Thank you for the helpful follow-up. Following your suggestion, we added an analysis on synthetic dataset and a qualitative case study on Adult, focusing on counterfactuals whose valid target recourse lies in the minority subgroup of high-income females.
> > >
> > > **A. Analysis on Synthetic Dataset**
> > >
> > > **A1 Minority Group.** We added a deviated subgroup to the moon dataset to represent a small but locally supported region. Results in Fig.1(https://jpst.it/4XAN7) show that CEs can still move into the subgroup, as long as they are locally reachable and not too costly.
> > >
> > > **A2 Diversity.** We use a query and sample 6 times during the inference phase. Fig.2(https://jpst.it/4XAN7) shows that the CEs corresponding to the query don't collapse to a single solution. For example, a single query can find CEs in two different regions if the cost constraint is satisfied.
> > >
> > > **B. Qualitative Study on Real Dataset**
> > >
> > > We further adopt a qualitative study on Adult to see whether the CEs can stay within the minority subgroup(high-income female ≈15%), instead of always drifting toward the dominant target mode(high-income male ≈85%).
> > > In the training set, samples with 'income >= 50K' accounted for ≈24%, of which only ≈15% were female. We first select samples labeled 'income<50K' from the test set with a male/female ratio of 50/50 as queries, and obtain CEs using DensityFlow. **Please note that the _'race'_ and _'sex'_ features are commonly immutable, but in this experiment, we made both operable.** The results of the male/female ratio are shown in Table 1. The generated CEs show a certain degree of shift towards the dominant modality, but a large proportion of the CEs results still remain in the female group.
> > >
> > > Table 1. Comparison of sex distribution between queries and CEs
> > > |            | Male    | Female   |
> > > |------------|---------|----------|
> > > | Queries    | 50.09%  | 49.91%|
> > > | Robust CEs | 56.98%  | 43.02%|
> > >
> > > To address your concerns, we filter out some cases for **female with high income** in Tables 2 and 3. The modified features (age, hours per week, occupation, etc.) are usually consistent with our perception of high income.
> > >
> > > Table 2. Case 1
> > >
> > > |                | age  | workclass | education     | marital.status      | occupation  | relationship | race  | sex    | capital.gain | capital.loss | hours.per.week | native.country | income |
> > > |----------------|------|-----------|---------------|--------------------|-------------|--------------|-------|--------|--------------|-------------|---------------|----------------|--------|
> > > | **Original**   | 20.0 | Private   | Some-college  | Never-married      | Sales       | Own-child    | White | Female | 0.0          | 0.0         | 20.0          | Other          | <=50K  |
> > > | **Counterfactual** | 35.0 | Private   | Bachelors      | Married-civ-spouse | Sales       | Wife         | White | Female | 0.0          | 0.0         | 34.0          | Other          | >50K   |
> > >
> > > Table 3. Case 2
> > >
> > > |                | age  | workclass   | education | marital.status      | occupation       | relationship | race  | sex    | capital.gain | capital.loss | hours.per.week | native.country | income |
> > > |----------------|------|-------------|-----------|---------------------|------------------|--------------|-------|--------|--------------|-------------|---------------|----------------|--------|
> > > | **Original**   | 19.0 | Private     | HS-grad   | Never-married       | Other-service    | Own-child    | White | Female | 0.0          | 0.0         | 12.0          | United-States  | <=50K  |
> > > | **Counterfactual** | 30.0 | Federal-gov  | Bachelors  | Married-civ-spouse  | Prof-specialty    | Wife         | White | Female | 2498.0       | 0.0         | 40.0          | United-States  | >50K   |
> > >
> > > We also add a distribution-level view. The KDE plot (https://jpst.it/4XC6a) for female hours-per-week shows that the CEs tend to move from low working hours to the 30–40 hours/week range. This is broadly consistent with a shift from **part-time work* toward more **regular/full-time work**, which makes the direction more plausible.

---

### Official Review · Reviewer_4hoC · 2026-03-11

**Soundness:** 3
**Presentation:** 3
**Significance:** 3
**Originality:** 3
**Overall Recommendation:** 4
**Confidence:** 4

**Summary:**

DensityFlow proposes a novel, density‑guided generative framework for producing robust counterfactual explanations (CEs) in the presence of model multiplicity.  The method couples a Neural‑ODE based continuous flow with a Noise‑Contrastive Estimation (NCE) discriminator that jointly learns class‑conditional density ratios and a surrogate classifier.  The learned “density score” steers the flow dynamics away from low‑support regions, thereby reducing model disagreement along the CE trajectory.  For black‑box settings a lightweight local distillation step is introduced that aligns a surrogate with the ensemble only within the visited trajectory, dramatically reducing query cost.  Extensive experiments on eight tabular datasets and four synthetic toy problems show that DensityFlow attains higher validity under model multiplicity than several state‑of‑the‑art baselines while keeping the transport cost comparable.  The authors also provide a thorough ablation study, sensitivity analysis of the NCE hyper‑parameters and visual diagnostics of trajectory agreement.

**Compliance With Llm Reviewing Policy:**

Affirmed.

**Final Justification:**

My concerns are resolved.

The three additional high-dim datasets, the honest failure-case analysis (conductivity/pH), and the disagreement-stratified experiment (Table 3) are convincing. The dopri5 comparison nicely addresses the discretization question. Two minor revision suggestions: briefly flag the causal-coupling failure mode in the main limitations section (not just the appendix), and add a sentence about the adaptive solver in the main text.

**Key Questions For Authors:**

1. Noise Distribution Choice

You use a uniform noise distribution to train the NCE discriminator. How sensitive are the results to this choice? Would a Gaussian or data‑dependent noise distribution improve density estimation, especially for highly non‑uniform tabular data?

2. Scalability to High‑Dimensional Data

The experiments are limited to ≤ 23 features. Neural ODEs can be computationally expensive in high dimensions, and density estimation via NCE may suffer from the curse of dimensionality. Have you tried DensityFlow on datasets with > 100 features, and if so, what were the challenges?

3. Hyperparameter Sensitivity

You report some analysis of the noise ratio τ and bound C, but the impact of other hyperparameters (e.g., λ₍cost₎, λ₍den₎, learning rates) on the trade‑off between validity and cost is not fully explored. Could you provide a more systematic sensitivity study or guidelines for selecting these values?

4. Handling Categorical Features

The paper focuses on tabular data but does not detail how categorical variables are embedded or transformed before feeding into the Neural ODE. Are one‑hot encodings used, or is an embedding layer learned? How does this affect density estimation and trajectory smoothness?

5. Local Distillation Robustness

The distillation loss uses a worst‑case aggregation over two surrogate proxies. How does this choice affect robustness when the ensemble exhibits highly divergent predictions? Would a more principled alignment (e.g., KL divergence to the ensemble’s predictive distribution) yield better gradients?

**Limitations:**

The authors briefly discuss that scaling the density‑score component to high‑dimensional modalities is non‑trivial and that future work will explore embedding‑level density estimation. However, a more thorough discussion of potential failure modes (e.g., under severe distribution shift, highly imbalanced classes, or very high‑dimensional data) would be valuable. Additionally, the impact of noisy labels on density estimation is not considered.

**Strengths And Weaknesses:**

# Soundness
The formulation is mathematically grounded; the NCE derivation is proven in Appendix A.1, and the Neural‑ODE dynamics are standard with an augmented state for kinetic energy. Experimental design is comprehensive, comparing against strong baselines on both synthetic and real data. Concerns:
- The impact of discretisation in the ODE solver on density guidance is not fully explored.
- The assumption that a uniform noise class yields an appropriate density threshold is only empirically justified.
- No formal analysis of the impact of the surrogate‑distillation step on optimisation stability.
- Convergence guarantees for the alternating training of surrogate and generator are not discussed.

# Presentation
The paper is well‑structured, with clear sectioning and a thorough related‑work survey. Figures illustrate key concepts (e.g., trajectory vs. density). However, some parts are dense: the description of the local distillation and the exact loss formulations (e.g., Eq. (6) vs. Eq. (7)) could be clarified with more intuition and a concise pseudo‑code block. The appendix is extensive but sometimes redundant with the main text.

# Significance
Robust counterfactuals are a timely topic in XAI. The proposed density‑guided ODE framework offers a principled way to avoid low‑confidence regions, which is valuable for high‑stakes decision making. The query‑efficient distillation also addresses a practical bottleneck in black‑box settings.

# Originality
Combining NCE‑based density estimation with Neural ODE dynamics for CE generation is novel. The local distillation strategy that restricts alignment to the trajectory is a creative solution to query cost. While each component (NCE, ODEs, distillation) has appeared elsewhere, their combination for robust CE is new.

---

> ### Author Rebuttal · Authors · 2026-03-30
>
> Thanks for your valuable review. As you suggested, we'll add convergence plots and pseudo-code, please refer to https://jpst.it/4XgXX. We now answer questions point by point below and will revise the paper accordingly.
>
> **Q1. Noise Distribution Choice**
>
> **A1.** Thanks for your question. Due to the finite samples, the choice of noise distribution directly influences the learned density score. As noted in Sec 4.1(Noise as a trust reference), we use uniform noise because it provides a simple yet consistent reference without introducing additional structural bias. By contrast, Gaussian/data-dependent noise concentrates much higher mass near its center and much lower mass in the periphery, making the estimator more sensitive to its parameterization.  To verify this, we designed a Gaussian variant that samples $x_{noise} \sim \mathcal N(\mu_{noise}, \Sigma_{noise})$ with $\Sigma_{noise}=s^2\big((1-\rho)\,\mathrm{Diag}(\Sigma_{train})+\rho\,\Sigma_{train}\big)+\varepsilon I$, where $\rho$ controls the correlation strength, and $s$ controls the overall scale. Table 1 shows that while Gaussian noise can be competitive if carefully tuned($s=1.4, \rho=0.9$), it is sensitive to its covariance structure and sampling balance. In contrast, uniform noise consistently performs well across various settings.
>
> Table 1. Noise and data-noise balance analysis on Spirals
> |Noise|Balance Ratio|Cost↓|Validity↑|
> |---|---|---|---|
> |Uniform|0.5|0.48|0.97|
> ||0.7|0.46|0.97|
> ||0.9|0.52|0.95|
> |Gaussian(s=1.4,$\rho$=0.9)|0.5|0.51|0.97|
> |Gaussian(s=1.0,$\rho$=0.6)|0.7|0.41|0.88|
> ||0.9|0.68|0.95|
>
> **Q2. Scalability to High-Dimensional **
>
> **A2.** Thanks for this insightful comment. Both density estimation and CEs face difficulties due to the 'curse of dimensionality'.
> This is an active research domain, and we alleviate it through two primary directions: (a) Using existing latent-based methods like β-VAE to project features onto lower-dim manifolds. (b) Then we note that not all features contribute equally; this aligns with the concept of feature selection (FS). We therefore designed an FS variant: we use LassoNet [2] to identify the Top-K features most relevant to the label, then feed only these features into DensityFlow. As shown in Table 2 on the Musk dataset (6598 samples, 169 features), this variant achieves strong validity at Top-5 with lower cost, Top-20 validity drops as some noise is added, suggesting that filtering out irrelevant dimensions effectively mitigates the dimensionality challenge.
>
> Table 2. Results on Musk dataset
>
> |Method|Cost↓|Validity↑|
> |---|---|---|
> |DensityFlow-FS(Top-5)|**8.57**|**0.86**|
> |*(Top-10)|8.73|0.85|
> |*(Top-20)|10.26|0.80|
>
> [1]Wu X, et.al. Local density estimation in high dimensions
>
> [2]Lemhadri I, et al. Lassonet: A neural network with feature sparsity
>
> **Q3. Hyperparameter Sensitivity**
>
> **A3.** We appreciate your suggestion. While sensitivity analysis for $\lambda_{cost}$ is provided in Appx. C.4, we have added Table 3 to evaluate $\lambda_{den}$. Results indicate that $\lambda_{den}$ improves validity by encouraging CEs to reside within stable regions, but at the expense of higher cost. The best trade-off is typically in the range of 0.05–0.2; larger values mainly increase cost. For the learning rate, standard settings such as $10^{-3}$ or $5 \times 10^{-4}$ proved stable.
>
> Table 3. Sensitivity to $\lambda_{den}$ with fixed $\lambda_{valid}=1.5$ and $\lambda_{cost}=0.3$ on Adult
> |$\lambda_\mathrm{den}$|Cost↓|Validity↑|
> |---|---|---|
> |0.00|**1.48**|0.86|
> |0.05|1.51|0.88|
> |0.10|1.52|0.88|
> |0.20|1.59|**0.90**|
> |0.40|1.84|0.89|
>
> **Q4. Handling Categorical Features**
>
> **A4.** Please refer to **A2** for Reviewer rEzf.
>
> **Q5. Local Distillation Robustness**
>
> **A5.** This is a really good question. We've also considered this issue: if we align directly, we can eliminate the overhead of querying the ensemble. However, boundaries under MM are inherently fuzzy and uncertain. When ensemble disagreement is large, accurate alignment becomes difficult. In contrast, our worst-case design can preserve this critical uncertainty, ensuring CE remains valid even under extreme proxy disagreement.
>
> Table 4 shows that the worst-case maintains higher validity. Nevertheless, the gradient efficiency of KL alignment is a distinct advantage we plan to explore deeply in future work.
>
> Table 4. Distillation target comparison on Adult
> |Target|Cost↓|Validity↑|
> |---|---|---|
> |Worst-case|1.59|**0.90**|
> |KL Div|**1.57**|0.86|
>
> **Q6. Limitations Discussion**
>
> **A6.** Thanks for your suggestion. We'll expand the limitations discussion to cover distribution shift, class imbalance, and very high-dimensional data. We also tested label noise by flipping a fraction $\eta \in \{0, 0.1, 0.2\}$ of training labels. Results in https://jpst.it/4Xh6S show a clear trend: more label noise lowers validity and slightly increases cost, because the estimated class-conditional support becomes less precise and the density guidance weakens.

---

> > ### Author Rebuttal · Reviewer_4hoC · 2026-04-05
> >
> > Thank you for the detailed rebuttal. Q1, Q4 and Q3 are resolved. The additional experiments are convincing. A few points remain:
> >
> > ## Q2 (Scalability).
> >
> > The LassoNet variant is pragmatic, but it assumes features important for classification are also appropriate for density-guided CE generation. Could you comment on when this might break down? Results on a single high-dimensional dataset are also somewhat limited, and the β-VAE direction remains unexplored experimentally.
> >
> > ## Q5 (Distillation Robustness).
> >
> > Table 4 is helpful, but Adult may not exhibit sufficiently high ensemble disagreement to stress-test the worst-case design. Do you have evidence that the advantage over KL divergence grows as disagreement increases?
> >
> > ## ODE Discretization.
> >
> > My original review raised the impact of solver step size on density guidance, which was not addressed. Does coarser stepping degrade the density score's effectiveness?

---

> > > ### Author Response · Authors · 2026-04-06
> > >
> > > Thanks for your feedback! We are happy to answer your remaining questions.
> > >
> > > ## A2 Scalability
> > > Yes, you are right. LassoNet variant assumes features predicting $P(Y|X)$ sufficiently overlap with the intrinsic manifold $P(X)$. It succeeds when most features are noise. Projecting data onto the most discriminative subspace lets Neural ODEs smoothly traverse the core manifold without high-dim disturbance.
> > >
> > > **Failure Case.** It fails when discarded ‘non-predictive’ features are physically/causally coupled with features the CE modifies. For example, in the 'Water Quality and Potability' dataset from Kaggle, a feature called 'conductivity' lacks predictive power for 'potability' and is filtered. If DensityFlow drastically alters 'pH', the original space requires a physical chain reaction in conductivity. Ignoring this yields OOD CEs (e.g., 'high pH, near-zero conductivity'). We'll formalize this in the appendix.
> > >
> > > **More high-dim datasets.** We add 3 high-dim datasets in Table 1. DensityFlow-FS maintains high validity compared with the recent baseline.
> > >
> > > Table 1. Comparison results of DensityFlow-FS (TOP-20) on 3 high-dim(>100) datasets
> > > |Method|Data(samples,dims)|Cost↓|Validity↑|
> > > |------|--------------------|-----|---------|
> > > |DensityFlow|madelon(2600,500)|3.59|0.87|
> > > ||gas(5491,128)|6.02|0.91|
> > > ||scene(2407,300)|13.3|0.82|
> > > |Argument|madelon|3.06|0.83|
> > > ||gas|6.67|0.92|
> > > ||scene|13.9|0.77|
> > >
> > > **Regarding the β-VAE.** For the experiments on β-VAE, please see **A3**(Reviewer yeSp). Table 2 shows that the latent variant(*-L) works well on gas, with a smaller cost, but drops sharply on scene/madelon. This is likely due to reconstruction errors caused by excessive smoothing of the latent space. We believe that 'how to counterfactual explanation on high-dim tabular data' can be considered a topic for future work.
> > >
> > > Table 2. Ambient space vs. latent space on high-dim datasets
> > > |Dataset|Method|Cost↓|Validity↑|
> > > |-------|------|-----|---------|
> > > |gas|DensityFlow-FS|6.02|0.91|
> > > ||*-L|5.74|0.90|
> > > |madelon|DensityFlow-FS|3.59|0.87|
> > > ||*-L|19.6|0.62|
> > > |scene|DensityFlow-FS|13.3|0.82|
> > > ||*-L|150.8|0.50|
> > >
> > > ## A5 Distillation Robustness
> > > Thanks for your follow-up question. Regarding CEs under MM, the 'high ensemble disagreement' you mentioned is an interesting but extreme scenario. Imagine that a well-tuned ensemble(XGB/MLP/SVM...) exhibits very high inconsistency for a dataset. The dataset is likely noisy, and the CEs given in such a scenario might be unreachable or lose diversity.
> > >
> > > **Analysis.** Assuming one extremely optimistic model gives 0.9, and another extremely pessimistic model gives 0.1, the KL divergence might guide the CE to a region with moderate confidence (0.5). The Worst-case design is a minimax optimization that forces the CE to satisfy the boundary of the most pessimistic model. Therefore, the greater the model divergence, the more pronounced the advantage of Worst-case becomes, as it prevents the CE from falling into the trap of being severely rejected by some models simply because of a 'high average score'.
> > >
> > > **Experiment.** To address your concerns, we designed an experiment on the Adult dataset, where we compared the results of the KL-div/worst-case in the following scenarios:
> > >
> > > a. Models in the ensemble are not tuned, i.e., default parameters(high-disagreement,  acc: 0.847±0.022)
> > >
> > > b. The worst model in the ensemble is tuned(medium-disagreement,  acc: 0.849±0.017)
> > >
> > > c. Models in the ensemble are fully tuned(low-disagreement,  acc: 0.858±0.001)
> > >
> > > Table 3 shows that the worst-case design is more effective than the KL-div as the ensemble divergence increases with a slightly higher cost.
> > >
> > > Table 3. Performance comparison under varying levels of ensemble disagreement
> > > |Scenario|Method|Cost↓|Validity↑|
> > > |--------|------|-----|---------|
> > > |high-disagreement|Worst-case|1.80|0.82|
> > > ||KL Div|1.70|0.77|
> > > |medium-disagreement|Worst-case|1.78|0.82|
> > > ||KL Div|1.73|0.80|
> > > |low-disagreement|Worst-case|1.59|0.90|
> > > ||KL Div|1.57|0.86|
> > >
> > > ## ODE Discretization
> > > We apologize for missing this question; the answer is '**yes**', a coarser step does indeed weaken the effect. However, the step is not a sensitive manual hyperparameter in our design. Table 4 shows that, while large, fixed steps may speed up, they can cause trajectories to deviate from robust regions, thus reducing validity. Small step sizes typically increase computation time but don't significantly change validity. In our implementation, we use an adaptive solver(**dopri5**), which automatically reduces the step size based on local error tolerance, thereby enabling finer sampling in regions with complex density gradients and preventing deviation from density guidance.
> > >
> > > Table 4. ODE step analysis on Adult
> > > |Solver|Step Size|Num Steps|Cost↓|Validity↑|Running Time(s)|
> > > |-------------|---------|---------|-----|---------|--------------|
> > > |Euler (Coarse)|0.5|2|3.58|0.65|8780.9|
> > > |*(Medium)|0.1|10|1.81|0.82|9917.2|
> > > |*(Small)|0.01|100|1.55|0.90|43827.3|
> > > |dopri5 (**Default**)|Adaptive|Auto|1.59|0.90|11136.5|

---

### Official Review · Reviewer_nkmS · 2026-03-12

**Soundness:** 3
**Presentation:** 3
**Significance:** 3
**Originality:** 3
**Overall Recommendation:** 5
**Confidence:** 2

**Summary:**

This paper addresses the problem of generating counterfactual explanations that is valid and robust (in the sense that they lie in the high-density regions). The authors propose DensityFlow that models counterfactual generation via a Neural ODE guided by a differentiable density score to steer trajectories toward high-density regions of the target class. The density score is derived from a Noise Contrastive Estimation formulation that augments the K semantic classes with a uniform noise class. For black-box settings, the paper introduces a local distillation that aligns the learnt surrogate with the target blackbox output only along the regions visited during generation to reduce query costs. The method is evaluated on synthetic and real-world tabular datasets against several baselines.

**Compliance With Llm Reviewing Policy:**

Affirmed.

**Final Justification:**

The rebuttal addresses my major concerns regarding clarity and presentation, so I increase my score to accept. But I focus mainly on clarity, presentation and logical flow of the manuscript (as I am not an expert in the field). I'll keep my confidence level to 2.

**Key Questions For Authors:**

As mentioned in Strengths And Weaknesses, I wonder if we need labeled data to train NCE or we can just use the predicted label from the blackbox model of interest? (I am not an expert in this field, so I may miss something here).

**Limitations:**

Yes

**Strengths And Weaknesses:**

Strengths:

- The paper is well-written and the core motivation is clear. I appreciate the well-made visualizations that help explain the core ideas and simplify the more complex parts.

- The method is logical with a logical result to back up the idea. The overall arguments are coherent.

- To my knowledge (I am not an expert in this field), the proposed method is original and tackles a relevant gap in the prior work

- The experiments are well designed and try to understand several important aspects of the proposed method

Weaknesses:
- The paper could be improved to make it more accessible to readers outside the field. For example, the authors state that they "restrict discussion to MM-based robustness", but model multiplicity is never formally defined. The paper uses the term throughout and it seems central to the problem formulation. If this is the core setup, it should be introduced properly.

- There are mathematical notations that are not defined, which can confuse the reader. I can point to two instances. First, in Equation 2, the validity constraint is written as E_M[h(x')] = y*, but M is not defined.  To me, this is important since Equation 2 is the central problem formulation. Second, in Equation 1, the trust region is defined using p_ref, but I cannot find where p_ref is defined. I would encourage the authors to carefully check all notations.

- It is unclear to me what exactly the data requirements of the method are. In Section 4.1, D_src is referred to as the source data, and Equation 3 draws (x, y) from D_src, suggesting it is labeled. But this requirement is not stated explicitly. As someone less familiar with this field, I wonder: why can we not use predicted labels from the black-box model in place of true labels, so that only unlabeled data is needed to train the NCE estimator? If ground-truth labels are necessary, this should be stated clearly.

---

> ### Author Rebuttal · Authors · 2026-03-29
>
> Thank you for the careful review. We appreciate your comments on clarity, notation, and data requirements, which were very helpful. We address these points below and will revise the paper accordingly.
>
> **Q1. Model multiplicity not formally defined.**
>
> **A1.** Thank you for pointing these out. We apologize that MM was not formally defined in the  problem setup. We'll make this explicit early in the Problem Statement as follows:
>
> **RCE under Model Multiplicity.**
> We study robust counterfactual explanations under model multiplicity (MM).  Similar to recent work, such as Jiang et al.[1], we consider a finite set of plausible predictors $\mathcal{M} = \(\{ h_j \}\)_{j=1}^{m}$,
> where $m$ denotes the number of candidate models, and each $h_j:\mathcal{X}\to\mathcal{Y}$ is a well-trained predictor for the same classification task, but different models may induce different decision boundaries. Accordingly, MM-based robustness asks whether a counterfactual remains valid across this model set, rather than only for one fixed predictor.
>
> [1]. Jiang et al. Recourse under Model Multiplicity via Argumentative Ensembling
>
> **Q2. Some notations are not defined.**
>
> **A2.** We apologize for these undefined symbols. In the revised manuscript, we explain $p_{\text{ref}}$ immediately after Eq.(1):   $p_{\text{ref}}$ denotes the reference distribution term, which serves as the background distribution for NCE-based trust-region construction. Thus, $\tau$ specifies the trust region as a relative density threshold, i.e., a fraction of the peak target-class density. Finally, we also carefully re-check the notation throughout the manuscript and ensure that all symbols are defined at first use.
> We also revise other undefined symbols. For example, in Eq.(8), where $\sigma(\cdot)$denotes the softmax probability for the target class.
>
> **Q3. Does NCE need labels?**
>
> **A3.** Thank you for this question. Yes, the NCE is trained with labeled training data, and we will state this requirement explicitly in the revision.
>
> In our framework, the role of NCE is to learn a density-related signal for the target class $y^*$. We want this signal to reflect where the data distribution itself provides support for that class, rather than the behavior of any single predictor. If we replaced ground-truth labels with the black-box model's prediction, the learned score would no longer reflect only the data support of the target class; it would also absorb the black-box model's own biases, errors, and boundary artifacts. In that case, the density signal would become model-dependent, which is not the role we want it to play.
>
> In the revision, we will make this distinction explicit: ground-truth labels are used to learn the class-support signal, while the black-box model is used to optimize and validate whether the generated counterfactual achieves the target prediction.

---

> > ### Author Rebuttal · Reviewer_nkmS · 2026-04-01
> >
> > Thank you for improving the presentation and address notational issues. I'll increase my score to accept. And my focus is more on clarity, presentation and logical flow (I am not an expert in this field) so I will keep my confidence at level 2.

---

> > > ### Author Response · Authors · 2026-04-02
> > >
> > > Thank you for the review and all the suggestions!  We will continue refining the manuscript for readability and flow.

---

### Official Review · Reviewer_rEzf · 2026-03-12

**Soundness:** 3
**Presentation:** 3
**Significance:** 2
**Originality:** 3
**Overall Recommendation:** 4
**Confidence:** 3

**Summary:**

This work proposes a method for generating counterfactual explanations under model multiplicity by encouraging solutions to lie in high-density regions. The approach follows a three-step pipeline. First, the original K-class classification problem is reformulated as a  (K+1)-class discrimination task, where the additional class represents a reference noise distribution. Second, an (augmented) Neural ODE is used to parameterise a continuous counterfactual trajectory starting from the query instance x, and 0 velocity. The flow is then optimised to minimise the trajectory length, guided by a density potential derived from the (K+1)-discriminator. This potential steers the trajectory toward a high-density region corresponding to a valid target class.

**Compliance With Llm Reviewing Policy:**

Affirmed.

**Key Questions For Authors:**

1. Why is a Neural ODE necessary instead of using simpler optimisation-based counterfactual methods?

2. How would the proposed approach handle categorical or discrete features?

3. What is the computational cost (runtime) of the full pipeline in practice, and how does it compare to the baselines?

4. Why is there no independent or external evaluation of counterfactual plausibility, particularly in comparison to the baselines?

**Limitations:**

Yes

**Strengths And Weaknesses:**

The paper is generally well written. Instead of directly learning the marginal density, the authors estimate a class-based (conditional) density, which is a clever design choice. This density estimator serves a dual purpose: it helps guide the counterfactual toward the desired target label while also encouraging solutions to lie in high-density regions of the data distribution. To my knowledge, using Neural ODEs in this context is also new.

The overall system appears somewhat heavy in terms of modelling complexity, but the approach itself seems methodologically sound. One potential drawback is the dependence on the noise ratio, which may be difficult to tune in practice; however, the authors do provide an ablation study analysing its effect.

I also have some reservations about how the method would extend to settings with categorical features. Additionally, while the authors report metrics such as cost and validity, the evaluation does not include an external or independent assessment of plausibility, which is also an important aspect of counterfactual explanations. The paper is generally well written. Instead of directly learning the marginal density, the authors estimate a class-based (conditional) density, which is a clever design choice. This density estimator serves a dual purpose: it helps guide the counterfactual toward the desired target label while also encouraging solutions to lie in high-density regions of the data distribution. To my knowledge, using Neural ODEs in this context is also relatively novel.

The overall system appears somewhat heavy in terms of modeling complexity, but the approach itself seems methodologically sound. One potential drawback is the dependence on the noise ratio, which may be difficult to tune in practice; however, the authors do provide an ablation study analyzing its effect.

I also have some reservations about how the method would extend to settings with categorical features. Additionally, while the authors report metrics such as cost and validity, the evaluation does not include an external or independent assessment of plausibility, which is also an important aspect of counterfactual explanations. Finally, the number of datasets used for evaluation is relatively small, which makes it difficult to assess how well the approach generalises to more realistic settings.

---

> ### Author Rebuttal · Authors · 2026-03-29
>
> Thank you for the careful review and constructive feedback. We‘ll answer each point below and will clarify these aspects in the revision.
>
> **Q1.Why Neural ODE?**
>
> **A1.** Thanks for raising this question. While simpler, static optimization methods (like gradient descent in the feature space) are effective for finding point-to-point perturbations, they strictly evaluate the start and end states. Consequently, they are **manifold-blind** between these two points, frequently crossing through low-density, highly uncertain regions of the feature space (as illustrated in Fig.3).
>
> Here, Neural ODE is adopted to host our specific framework designs because we formulate RCE as a continuous, density-constrained optimal transport problem. The Neural ODE backbone naturally integrates our three core contributions in ways that static optimization cannot: (a) density-ratio estimation to guide the path direction;(b) state-augmented transport cost along the trajectory; (c) trajectory-aware local distillation for aligning the local black-box boundary.
>
> As you and Reviewer yeSp pointed out, Neural ODE isn't the only choice, but it is a simple yet effective design to achieve our needs. Other latent or generative parameterizations could also be used in our framework with density-guided enhancement.
>
> **Q2.How to deal with categorical features?**
>
> **A2.** In our design, we do account for categorical features, but since handling them is not a core contribution of our work, we did not elaborate on it in the method section. We did used **datasets with categorical features** (e.g., Adult & Compas). Following the strategy used in TabRep [1], we encode each categorical feature by projecting it onto the two-dimensional unit circle, as implemented in our source code. Specifically, for a feature with $K$ categories, the category $k_i$ is represented as:
> $\mathbf{E}(k_i) = \left[ \cos\left(\frac{2\pi k_i}{K}\right), \sin\left(\frac{2\pi k_i}{K}\right) \right]$
>
> Compared with onehot encoding, this representation keeps the dimension low and provides a continuous geometry. We also conducted an experiment to compare the two encoding methods in Table 1, where the unit-circle encoding is more favorable on both datasets. We apologize for not clarifying this in the main text and will update our manuscript.
>
> Table 1. Categorical encoding comparison, DensityFlow \|Onehot-Variant
> |Dataset | Cost(↓)  | Validity(↑) |
> |---|---|---|
> |Adult   | **1.55**\|1.68 | **0.90**\|0.87 |
> |Compas  | **1.17**\|1.19 | **0.73**\|0.72 |
>
> [1] Si J et al. TabRep: Training Tabular Diffusion Models with a Simple and Effective Continuous Representation
>
> **Q3. Running time comparison?**
>
> **A3.** Thank you for the suggestion. Here we report the end-to-end runtime of the full pipeline. Since the compared baselines are quite different in nature(e.g., search-based, generation-based, optimization-based), a unified theoretical complexity comparison would be less informative. The results in Table 2 show that DensityFlow is slightly more expensive than some simpler baselines, which is expected given the additional modules. However, this cost is accompanied by substantially better robustness under MM.
>
> Table 2. Running time comparison (in seconds)
> | Dataset/Method | Product_mip | CeFlow | Batarce | Argument | DensityFlow |
> |---|---:|---:|---:|---:|---:|
> | Spirals | 704.7 | 394.6 | 747.9 | 205.7 | 655.6 |
> | HELOC   | 335.1 | 881.5 | 427.3 | 647.7 | 734.0 |
>
> **Q4.  Independent or external evaluation.**
>
> **A4.** Thanks for the suggestion, it will definitely strengthen our work. Following your suggestions, we use external plausibility metrics, LOF and Isolation Forest for evaluation[2]. Table 3 shows our work still maintains competitive plausibility under both metrics, consistent with our main conclusion from other metrics. We'll update our results.
>
>
> Table 3.  Performance comparison on two plausibility metrics
> |Dataset|Method|LOF(↓) \| ISOFOREST(↓)|
> |---|---|---|
> |Spirals|DensityFlow|**1.07\|0.48**|
> ||CeFlow|1.25\|0.59|
> ||betarce|1.13\|0.49|
> ||argument|1.16\|0.50|
> ||product_mip|1.29\|0.53|
> |HELOC|DensityFlow|**1.04**\|**0.42**|
> ||CeFlow|2.59\|0.48|
> ||betarce|3.17\|0.57|
> ||argument|1.07\|**0.42**|
> ||product_mip|1.54\|0.44|
>
> [2]Leofante F, et.al. Counterfactual explanations and model multiplicity: a relational verification view
>
> **Q5. The datasets are relatively small**
>
> **A5.**  We choose these datasets to align with recent works, e.g., Argument & CeFlow, to have fair comparisons. We also utilized three synthetic datasets (circles, spirals, and checkerboards) to test performance on complex manifolds. As suggested, we added the Diabetes(81413,  36) and German Credit(1000, 20) datasets to represent the medical and financial fields. Results in https://jpst.it/4W_gB show that DensityFlow maintains the advantage in validity and cost.
>
> We can also apply our work to high-dimensional datasets. Please refer to our **A4** for Reviewer 4hoC.

---

> > ### Author Rebuttal · Reviewer_rEzf · 2026-04-02
> >
> > The authors' responses adequately address the points I raised.

---

> > > ### Author Response · Authors · 2026-04-04
> > >
> > > We are glad that our responses have addressed your concerns, thanks for your detailed review and all suggestions.

---

### Decision · Program_Chairs · 2026-04-30

**Decision:**

Accept (regular)

**Comment:**

**Summary and Decision**
While reviewers noted minor concerns regarding the complexity of modeling and clarity in places, the general consensus is that the paper makes a timely and significant contribution to counterfactual explanations.

The reviewers specifically highlighted the following strengths:
* The significance of robust counterfactuals in XAI.
* The novelty of NCE-based density estimation with Neural ODE dynamics for CE generation.
* The paper is well-written.

**Rebuttal and Discussion**
During the discussion phase, the authors addressed issues concerning clarity, related work, and questions about method design by clarifying notation, expanding citations, and providing additional experiments, which satisfied the reviewers. Given the originality and soundness of the work, the Area Chair recommends acceptance.

**Final Instructions to Authors**
Please ensure that the promised revisions from the rebuttal—specifically the title change and related clarifications—are incorporated into the camera-ready version. Also, reviewers specifically suggested the following in discussions:

* "Briefly flag the causal-coupling failure mode in the main limitations section (not just the appendix), and add a sentence about the adaptive solver in the main text."
* "I would encourage you to clearly state in the paper that you in fact, are not sacrificing diversity completely and the report these qualitative results and the diversity plot at least in the appendix of the camera-ready."